# Simulation of 3D Body Shapes for Pregnant and Postpartum Women

**DOI:** 10.3390/s22052036

**Published:** 2022-03-05

**Authors:** Chanjira Sinthanayothin, Piyanut Xuto, Wisarut Bholsithi, Duangrat Gansawat, Nonlapas Wongwaen, Nantaporn Ratisoontorn, Parut Bunporn, Supiya Charoensiriwath

**Affiliations:** 1National Electronics and Computer Technology Center, National Science and Technology Development Agency, Pathum Thani 12120, Thailand; wisarut.bholsithi@nectec.or.th (W.B.); duangrat.gansawat@nectec.or.th (D.G.); nonlapas.wongwaen@nectec.or.th (N.W.); nantaporn.ratisoontorn@nectec.or.th (N.R.); parut.bunporn@nectec.or.th (P.B.); supiya.charoensiriwath@nectec.or.th (S.C.); 2Faculty of Nursing, Chiang Mai University, Chiang Mai 50200, Thailand; piyanut.x@cmu.ac.th

**Keywords:** 3D body shapes, body weights and measures, postpartum period, pregnancy period, anthropometry

## Abstract

Several studies have reported that pre-pregnant women’s body mass index (BMI) affects women’s weight gain with complications during pregnancy and the postpartum weight retention. It is important to control the BMI before, during and after pregnancy. Our objectives are to develop a technique that can compute and visualize 3D body shapes of women during pregnancy and postpartum in various gestational ages, BMI, and postpartum durations. Body changes data from 98 pregnant and 83 postpartum women were collected, tracked for six months, and analyzed to create 3D model shapes. This study allows users to simulate their 3D body shapes in real-time and online, based on weight, height, and gestational age, using multiple linear regression and morphing techniques. To evaluate the results, precision tests were performed on simulated 3D pregnant and postpartum women’s shapes. Additionally, a satisfaction test on the application was conducted on new 149 mothers. The accuracy of the simulation was tested on 75 pregnant and 74 postpartum volunteers in terms of relationships between statistical calculation, simulated 3D models and actual tape measurement of chest, waist, hip, and inseam. Our results can predict accurately the body proportions of pregnant and postpartum women.

## 1. Introduction

Obesity during pregnancy is a serious health problem for women. Worldwide, obstetricians and midwives have confronted increasing obesity among pregnant women [1,2]. Reports [3,4,5] showed that women with a pre-pregnant body mass index (BMI) of either overweight or obese levels are at risk of developing diabetes during pregnancy compared to women with normal pre-pregnant BMI, even after taking the weight gains during a normal pregnancy into account. Women with diabetes during pregnancy tend to have high blood pressure, which can lead to abdominal surgery and premature birth [6]. Therefore, it is important for all women who are planning to become pregnant to control a proper weight before and during pregnancy using multi-faceted interventions throughout the reproductive years as a part of a long-term follow up and behavioral interventions to minimize pregnancy weight gain [7]. The BMI before pregnancy affects not only the weight gain during pregnancy, but also the postpartum weight retention [8,9]. It was reported that if a woman in the postpartum period was unable to regulate her weight to her pre-pregnant weight within six months, postpartum weight retention could predict future weight gain and long-term obesity [10]. Another study suggested that the BMI of a woman of more than six months postpartum would indicate the retaining of extra body fluids produced during pregnancy, as well as extra fat during the first six months postpartum [11]. Sui Z. et al. [12] reported a statistically significant indication that women with a high degree of body image dissatisfaction were more likely to have higher gestational weight gain. Hill B. et al. [13] also reported that the timing of pregnancy and body attitudes could predict gestational weight gain (GWG). The findings suggested that lower attractiveness in early-to-middle pregnancy was associated with higher GWG [13]. Therefore, in this work, the 3D body shape simulation of women before, during, and after pregnancy has been developed with an expectation that women will not misestimate their BMI to prevent being overweight during pregnancy and the postpartum period. A 3D body simulation may encourage women to prevent overweight behaviors before and throughout pregnancy to maintain good health in the long run. Having 3D models in three stages of before, during, and after pregnancy to compare, is one of the ways to motivate women to develop long-term healthy behaviors. 

There are some research reports that look at correlations between body shapes and BMI for women [14,15,16]. Nevertheless, there is no physical simulation for the shape proportion of pregnant and postpartum woman subjects, as they are a population group vulnerable to the use of the 3D body scanner for data collection. The word “vulnerable” is in the context of human research protections. Pregnant women are considered vulnerable due to the involvement of the fetus that may be affected by the research and the fetus cannot give consent [17]. It was not possible to collect the body shape data using the 3D body scanner in the study. The reason for this is that most pregnant women are concerned about the safety of 3D body scanners and have questions about the potential consequences of the use of the scanner at all stages of pregnancy [18]. It is difficult to obtain an approval for applications for research projects involving human subjects from the Institutional Review Board (IRB), and especially when asking for consent forms for research subjects [17]. We also needed pregnant and postpartum women to measure their own body circumferences at home every four weeks during pregnancy and postpartum.

Therefore, it was necessary to collect data of various shapes using a tape measurement of pregnant and postpartum women by forward tracking six months before and after giving birth. Then data were analyzed and processed to create a simulation of 3D modeling of pregnant and postpartum women based on data of non-pregnant female shapes from SizeThailand [19,20]. Although there is a web application with a non-pregnant female simulation, including Body Visualizer [21] developed by Black and Broscaru [22] from Max Planck Gesellschaft, which published as a part of US Patent Application [23] and became US Patent [24]. The process mentioned in the patent has been in use for studying female sensitivity to changes in their perceived weight by altering the body mass index (BMI) of the participants’ personalized avatars to deal with body perception [25]. Furthermore, the process has also been in use for body size estimation in females varying in BMI as a measure to deal with rising cases of Anorexia patients [26]. In addition, the process uses the virtual caliper for the accurate 3D body measurements [27]. 

Body Visualizer has used the dataset based on SizeUSA, American and European Surface Anthropometry Resource (CAESAR) as the basis for creating 3D body shapes visualization [28,29]. It is a visualization tool for a parametric 3D body model that provides metrically accurate anthropomorphic measurements based on laser scans of thousands of people from different ethnicities. However, it is still lacking the 3D body models for pregnant and postpartum women, especially for Asian women. Therefore, in this study we focused on simulating the 3D shape of pregnant and postpartum women for Thais. In this article, 3D body shape simulation of non-pregnant, pregnant, and postpartum womens’ shape with body proportions are predicted in real-time and online from weight, height and gestational age. A real-time prediction in our study is a service that provides the predictions via an HTTP call to simulate the 3D shape of pregnant and postpartum women via the web browser after the users input their data. The body shape proportion of pregnancy and postpartum women were analyzed using the linear regression of 587 pregnancy data and 503 postpartum data. The simulations of pregnant and postpartum women were further modified based on non-pregnant simulations from our previous study, which was analyzed from the SizeThailand database with 6767 females’ data [19,20]. SizeThailand is a national sizing surveys project that includes 13,442 adults, both males, and females across Thailand at various ages [30].

Application Z-Size Ladies [31], described in this paper, was intended to collect user data in the form of BMI timeline and simulate the 3D female body shape for non-pregnant, pregnant women and postpartum women. Z-Size Ladies application [31] is a tool that helps pregnant and postpartum women to simulate their 3D-body shapes. The aim of developing the Z-Size Ladies application is to be a tool that can create a precise online 3D model for non-pregnant, pregnant, and postpartum women in real-time with a simple set of input data of their weight, height, and gestational age. The tool was validated by several linear regression studies and users’ survey. The simulation of the body shape from body measurements can be considered as a low-cost alternative to full-body 3D scanning [32]. Furthermore, this application can be applied to provide online clothing services. The users only put in their weight, height, and gestational age; then they will know their body shape proportions. More supporting information about Z-Size Ladies can be showed/downloaded in Appendix A.

## 2. Literature Reviews

There are some relevant studies on the prediction of 3D body shape during pregnancy using multiple 3D body scans with a purpose of setting the standard sizing chart for maternity wear that addresses the changes throughout pregnancy [33]. Vaughan et al. [34] matched personal weight, height and age with the overall body shapes taken from Magnetic Resonance Imaging (MRI) images to create 3D adjustable parametric human body models using OpenGL with 3D mesh deformation along with Artificial Neural Networks (ANNs) trained and assessed with the clinical data of 23,088 patients, including pregnant and postpartum patients from the National Health and Nutrition Examination Survey (NHANES) data from 1999 to 2012. The ANNs used in their study managed to predict the anthropometric measurements with the following margins of error including subscapular skinfold thickness within 3.54 mm, waist circumference 3.92 cm, thigh circumference 2.00 cm, arm circumference 1.21 cm, calf circumference 1.40 cm, and triceps skinfold thickness of 3.43 mm. An alternative regression analysis method gave overall predictions slightly less accurate for subscapular skinfold thickness within 3.75 mm, waist circumference 3.84 cm, thigh circumference 2.16 cm, arm circumference 1.34 cm, calf circumference 1.46 cm, and triceps skinfold thickness 3.89 mm. The results showed a parametric model of the patient’s body shape and ligament thickness using OpenGL and adjusted by 3D mesh deformation. However, the 3D image that resulted from the mesh deformation looked unrealistic, despite the accurate anthropometric measurements.

Haddox et al. [35] created a musculoskeletal model of a pregnant woman to simulate the changes in segmental mass and inertia distribution. It included a case of changing breast size during pregnancy. That caused pregnant ladies to fall due to the changes on the centers of the upper trunks, pelvis regions, and torso centers along with lumbar curvatures. They used datasets from 25 pregnant Caucasian ladies in six sessions and postpartum women obtained from US Air Forces Research Lab as models having BMI before pregnancy between 18.9 to 26 kg/m^2^. That was substantially lower than the average BMI of American women at 26.5 kg/m^2^. 

Ponnalagu et al. [36] pointed out that waist circumference (WC) is a simpler anthropometric measurement that has strong association with an individual’s metabolic risk level. BMI alone is not adequate since Asians have a high tendency to deposit fat at the viscera compared with their European counterparts. This explains why Asians have a higher fat percentage than Europeans despite having the same BMI. Furthermore, high waist circumference increased the risks of developing hypertension, type 2 diabetes mellitus, hypercholesterolemia, joint pain, low back pain, and hyperuricemia as mentioned in the paper by Darsini et al. [37].

Han et al. [38] investigated the cut-off points of body mass index (BMI) and waist circumference (WC) for gestational diabetes mellitus (GDM) and interactions between high BMI and high WC on the risk of GDM. They collected the data during 2010 to 2012 from 17,803 Chinese pregnant women from Tainjin who were at 4–12 week gestation. The results showed that higher than 22.5 kg/m^2^ BMI and higher than 78.5 cm WC were the cut-off points for gestational diabetes mellitus (GDM).

Jacobson et al. [39] invented Electronic Monitoring of Mom’s Schedule (eMOMS™) for monitoring improved postpartum weight, blood sugars, and breastfeeding among high BMI women who had BMI between 25 to 35 kg/m^2^. It offered an interactive communication between patients and physicians via Facebook, FaceTime and Skype. However, this invention focused only on weight rather than taking other anthropometric variables into account to help postpartum mothers to keep other physical factors of the body in check. 

Ha et al. [40] conducted studies on postpartum weight retention in relation to gestational weight gain and pre-pregnancy BMI due to the rising cases of maternal overweight and obesity in Vietnam. They studied 2030 pregnant women recruited from three cities in Vietnam who were 24–28 weeks of gestation for the analyses on gestational weight gain (GWG). In addition, they followed 1666 mothers for 12 months after delivery for the analyses on 12-month postpartum weight retention (PPWR). They recorded all pre-pregnant BMI. The results showed that both pre-pregnancy BMI and GWG were significantly associated with PPWR since those pregnancies with underweight before pregnancy and excessive GWG contributed to greater weight retention twelve months after giving birth. The measures to prevent postpartum maternal obesity should target at risk women who are underweight or overweight at the first antenatal visit and control their weight gain during the course of pregnancy. 

Nagpal et al. [41] carried out analyses on postpartum weight retention (PPWR) on 150 participants while taking anthropometric variables other than BMI and weight into account, such as waist circumference, hip circumference, and waist hip ratio. The results showed the postpartum weight retention was associated with the anthropometric measurements including waist circumference, hip circumference and waist-hip ratio. Increasing waist circumference and hip circumference could be applied to make the risk assessment for developing non-communicable diseases (NCD), such as gestation diabetes, which have been rising during the post-partum period.

## 3. Methodology

The Faculty of Nursing at Chiang Mai University and Maharaj Nakorn Chiang Mai Hospital approved the ethical authorization document for this study for data collection of pregnant and postpartum mothers with the following objectives:

(A) To collect six-month forward tracking data on body weight, body circumference, chest, waist, hip, upper arms and thighs during the pregnancy and postpartum period;

(B) To carry out an accuracy test of the Z-Size Ladies program in terms of 3D shape simulation compared with women during pregnancy and postpartum period.

This study was a ‘prospective study’ on volunteering pregnant women and postpartum women. The samples were women from antenatal care, who used postnatal services, and took their babies for vaccination at secondary and tertiary hospitals. A sample size of 98 pregnant and 83 postpartum women was analyzed for calculating model shapes of the pregnant and postpartum women. For the 3D simulation testing, a new group of 75 pregnant women and 74 postpartum women were compared in body shape proportion. The 3D body shape simulation of pregnant and postpartum women is based on the simulation of the non-pregnant female shape simulation studied in Sinthanayothin et al. [42]. 

The data collection started at 12–16 weeks of pregnancy and zero weeks of postpartum. The research assistant measured the body circumferences of chest, waist, hip, thighs (left/right), and upper arms (left/right) and explained to the volunteers how to measure their body size by themselves. The measurement was delicate, therefore the interrater agreement was essential that the participants had to hold the measuring tape in the correct position and not too tight. The measurement values could vary approximately ±2 cm. The measurements were taken at home by pregnant/postpartum women subjects with the assistance of someone at home. The measurement was taken every four weeks. There would be a reminder notice from the research assistant when the schedule was approaching. Three measurements were taken at each position and the median value was recorded for each position. The volunteers sent their measured data to the research team via mobile LINE application each time they measured their body shape.

### 3.1. The Simulation of the Female Shape in Three Dimensions (Z-Size Ladies)

The 3D simulation results for non-pregnant female bodies in various weights and heights are shown in Figure 1 [42].

### 3.2. The Correlation Analysis of Pregnant Women’s Body Proportion Using Multiple Linear Regression of the 587 Data Collected from 98 Pregnant Women

The data from 98 pregnant women volunteers were collected and analyzed. Information of all pregnant volunteers is shown in Table 1. The data were 587 sets in total. Each data contained woman’s age, pre-pregnancy weight, height, gestational age, weight gain during pregnancy, inseam (measure once at 12–16 week pregnancy), and body circumference measurements: chest, waist, hip, upper arm (left/right), and thigh (left/right). The data used for this study were from 94 women who were 12-week gestation; 98 of 16-week gestation; 91 of 20-week gestation; 82 of 24-week gestation; 79 of 28-week gestation; 78 of 32-week gestation and 65 of 36-week gestation, a total of 587 sets.

Wendland et al. [43] investigated the relationship between waist circumference and obesity-related pregnancy. The variables used in the correlation analysis were age, height, gravida, gestational age, uterine height, gestational BMI, and pre-pregnancy BMI. Similar work by Ricalde et al. [44] reported that some postpartum women’s anthropometric was related to birth weight. Therefore, in this pregnancy study, the relationships between the body shape proportion and variables of pregnancy such as woman’s age, weight, height, gestational age, weight gain during pregnancy were analyzed using multiple linear regression, which could be calculated in Excel [45].

The correlation indicates the relationship between the body shape proportion and variables of pregnancy such as woman’s age, weight, height, gestational age, weight gain during pregnancy, and so on as shown in Equation (1), where Value is the proportion of pregnancy woman’s body: Chest, Waist, Hip, Upper Arm, Thigh, respectively. *Y_w_* = Woman’s age (Default is set to 30 in case age is unknown), *W_pp_* = Pre pregnancy weight (Kg), *H_w_* = Women’s Height (cm), Gravida = Number of pregnancies (The default is set to 1, when pregnant for the first time), *Wk_p_* = Gestational age or Pregnant week (Weeks) and *W_g_* = Weight gain during pregnancy (Kg), respectively.
(1)$$Value=\left(A\times Y_{w}\right)+\left(B\times W_{pp}\right)+\left(C\times H_{w}\right)+\left(D\times Gr\right)+\left(E\times Wk_{p}\right)+\left(F\times W_{g}\right)+G$$

Although there is no direct factor of BMI categories in our correlation analysis, the body proportions, chest, waist, hip, thigh, and upper arm circumferences were calculated using a multiple linear regression method based on 587 data collected from 98 pregnant women. However, when a user wanted to predict her 3D pregnancy shape at other gestation ages using the web app (Z-Size Ladies), weight gain during pregnancy was unknown. Therefore, weight gain during pregnancy (*W_g_*) would be predicted from pre-pregnancy BMI as shown in Table 2 based on the Institute of Medicine (IOM), 2009 [46].

### 3.3. Simulation of Pregnant Women in 3D

The real-time visualization of 3D morphing of pregnant and postpartum female body shapes on the online Z-Size Ladies web application was implemented using the three.js library [47] incorporated with HTML5, JavaScript, and CSS for client-side development. Python, flask, and MySQL were employed for the server-side. Three.js was used as it was a cross-browser JavaScript library to ease the process of creating and displaying real-time 3D computer graphics and animation in the web browser.

A 3D model of a pregnant woman was created from a pregnant thin avatar shown in Figure 2 using a female model from Turbosquid [48,49] and combined with other avatars shown in Figure 3 for simulating a pregnancy figure by morphing technique. 

The pregnant avatar refers to the pregnant women’s simulation based on our previous studies [42]. The pregnant woman simulation is a combination of a woman shape and a pregnant shape. As the non-pregnant female simulation is a combination of a thin avatar (Figure 3A) and other avatar shapes (including Figure 3B big breast, (C) big waist, (D) big hip, (E) tall avatar, (F) long legs), so to simulate a pregnant woman, a pregnant thin avatar (Figure 2 or Figure 3G) is added. 

TurboSquid is a digital media company that sells 3D models used in 3D graphics to a variety of industries, including computer games, architecture, and interactive training [48,49].

Details of creating a 3D non-pregnant female shape by combining thin, big breast, big waist, big hip, tall and long legs avatars can be found in the article by Sinthanayothin et al. [42]. The idea of utilizing a combination of the avatar bodies for 3D shape simulation came from the morphing technique [50]. Morphing is a geometric interpolation technique, which mixed different characteristics of the objects. The body shape simulation that adjusted only a specific part was a challenge. For example, chest or hip circumference could be set bigger or smaller with the least impact on the waist and others. Therefore, our team designed the avatars in different ways to combine the shape of the body and to be able to adjust the size of specific parts as needed. Therefore, the 3D non-pregnant female shape was created by combining thin, big breast, big waist, big hip, tall and long legs avatars using the morphing technique to make it easier to adjust only a specific part of the body.

For simulating a pregnant body shape, the morphing technique was applied as shown in Equation (2): (2)$$P_{i}=(1-\overset{5}{\underset{i=0}{\sum }}K_{i})\times A_{i}+\left(K_{0}\times X_{i}\right)+\left(K_{1}\times B_{i}\right)+\left(K_{2}\times C_{i}\right)+\left(K_{3}\times D_{i}\right)+\left(K_{4}\times E_{i}\right)+\left(K_{5}\times F_{i}\right)$$
where *X_i_* is pregnant thin avatar, *A_i_*–*F_i_* are avatars with thin, big breast, big waist, big hip, tall and long legs, respectively. 

The simulation of a non-pregnant body shape from our previous study [42] showed that the variables *K*_1_–*K*_5_ depended upon the BMI values. Therefore, a similar experiment was performed in this study by testing 30 pregnant female subjects whose 3D data were simulated using Equation (2) in comparison with the statical measurement from Equation (1). 

In our experiment, the *K*_0_–*K*_5_ values are the sum between *K*_00_–*K*_05_ and the corresponding values between *Alp*_0_–*Alp*_5_ shown in Equation (3a,b). Where *K*_00_–*K*_05_ are depended on the values of *Chest*, *Waist*, *Hip*, *Height* and *Inseam* as shown in Equation (3c,d) respectively.
(3a)$$K_{0}=K_{00}+Alp_{0},\;K_{1}=K_{01}+Alp_{1},\;K_{2}=K_{02}+Alp_{2}$$
(3b)$$K_{3}=K_{03}+Alp_{3},\;K_{4}=K_{04}+Alp_{4},\;K_{5}=K_{05}+Alp_{5}$$
(3c)$$\mathrm{Where}\;K_{00}=0,\;K_{01}=\frac{\left(Chest-57\right)}{\left(200-57\right)},\;K_{02}=\frac{\left(Waist-40\right)}{\left(160-40\right)},$$
(3d)$$K_{03}=\frac{\left(Hip-68\right)}{\left(180-68\right)},\;K_{04}=\frac{1.36\times \left(Height-165\right)}{\left(199-165\right)},\;K_{05}=\frac{\left(Inseam-78\right)}{\left(120-48\right)}\;$$

The simulations of pregnant and postpartum women were further modified based on data from non-pregnant females from our previous study [42]. The constant values of Equation (3) were derived from the size of the designed avatars as mentioned in [42]. “The thin avatar was used as a default or initial model with the minimum values of chest, waist, and hip of approximately 57, 40, and 68 cm, respectively. The avatar with the big breast was applied to adjust the size of the chest values. The chest circumference of the avatar with the big breast was set as maximum chest values of approximately 200 cm. Similarly, for the avatar with the large waist and with the big hip, the size of the waist and hip of these avatars were set as maximum values of approximately 160 and 180 cm, respectively. For inseam, the default value for the thin avatar was approximately 73 cm. In this work, the minimum and maximum values of the inseam have been set to 48 and 120 cm, respectively. The last avatar (tall avatar) with the height of 200 cm is set to be the maximum height value.”

The values of Chest, Waist, Hip, and Inseam, which are the values that defined *K*00–*K*05, were calculated from the non-pregnant female body shape according to the article by Sinthanayothin et al. [42], which can be expressed by linear equations shown in Equation (4a–d) respectively.
(4a)Chest=0.872260×Weight−0.437949×Height+110.131573
(4b)Waist=0.931735×Weight−0.497702×Height+104.780946
(4c)Hip=0.729978×Weight−0.152380×Height+78.526838
(4d)Inseam=−0.059182×Weight+0.547734×Height−14.226815

From our experiments of cross-sectioning and measuring the circumference of 3D simulation figures, the *Alp*_0_–*Alp*_5_ were functions of the BMI, which could be calculated as the following polynomial equations. The quadratic functions derived from second-order polynomial regression and parameters from the experiment were performed to obtain a 3D pregnant woman model that was the closest to the calculated statistical value as shown in Equation (5a–f).
(5a)$$Alp_{0}=\left(Alp_{01}\times BMI\times BMI\right)+\left(Alp_{02}\times BMI\right)+Alp_{03}\;$$
(5b)$$Alp_{1}=\left(Alp_{11}\times BMI\times BMI\right)+\left(Alp_{12}\times BMI\right)+Alp_{13}\;$$
(5c)$$Alp_{2}=\left(Alp_{21}\times BMI\times BMI\right)+\left(Alp_{22}\times BMI\right)+Alp_{23}\;$$
(5d)$$Alp_{3}=\left(Alp_{31}\times BMI\times BMI\right)+\left(Alp_{32}\times BMI\right)+Alp_{33}\;$$
(5e)$$Alp_{4}=\left(Alp_{41}\times BMI\times BMI\right)+\left(Alp_{42}\times BMI\right)+Alp_{43}\;$$
(5f)$$Alp_{5}=\left(Alp_{51}\times BMI\times BMI\right)+\left(Alp_{52}\times BMI\right)+Alp_{53}\;$$

*Alp_XY_* is a variable that depends on the gestational age (*Wkp*), so it can be written as a quadratic function shown in Equation (6a).
(6a)$$Alp_{XY}=\left(Alp_{XYC}\times Wkp\times Wkp\right)+\left(Alp_{XYB}\times Wkp\right)+Alp_{XYA}$$
where $Alp_{XYA}$, $Alp_{XYB}$ and $Alp_{XYC}$ are constants calculated by polynomial fitting shown in Table 3. These values were applied to the morphing equations to obtain 3D pregnant women model closed to the calculated statistical value, as shown in Equation (1).

If *Wkp* (the gestational age) is less than 12 weeks, *Alp*_0_ can be calculated as shown in Equation (6b).
*Alp*_0_ = *Alp*_0_ × Wkp/12(6b)

### 3.4. The Correlation Analysis of Postpartum Women’s Body Proportion Using Multiple Linear Regression of the 503 Data Collected from 83 Postpartum Women

The data from 83 postpartum women volunteers were collected and analyzed. Information of all postpartum volunteers was shown in Table 4. The data were 503 sets in total. Each data contains the woman’s age, pre-pregnancy weight, height, gravida, baby weight, postpartum week, postpartum weight, inseam (measure once at zero weeks of postpartum), and body circumference measurements: chest, waist, hip, upper arm (left/right), and thigh (left/right). The data used for this study were from 81 women of 0-week postpartum; 76 of 4-week postpartum; 73 of 8-week postpartum; 72 of 12-week postpartum; 72 of 16-week postpartum; 70 of 20-week postpartum and 59 of 24-week postpartum, a total of 503 sets.

The correlation indicates the relationship between the body shape proportion and variables of postpartum such as woman’s age, pre-pregnancy weight, height, gravida, baby weight, postpartum week and postpartum weight, as shown in Equation (7), which is similar to Equation (1), however, with postpartum parameters. The values are the proportion of a postpartum woman’s body: Chest, Waist, Hip, Upper Arm, Thigh, respectively. *Y_w_* = Woman’s age (Default is set to 30 in case age is unknown), *W_pp_* = Pre pregnancy weight (Kg), *H_w_* = Height (cm), *Gr* = Number of pregnancies (The default is set to one, when pregnant for the first time), *W_b_* = Baby weight in Kg (The default is set to three in case baby weight is unknown), *Wk_ppt_* = Postpartum week (Weeks) and *W_ppt_* = Postpartum weight (Kg), respectively.
(7)$$\begin{array}{c} Value=\left(A\times Y_{w}\right)+\left(B\times W_{pp}\right)+\left(C\times H_{w}\right)+\left(D\times Gr\right)+\left(E\times W_{b}\right)\\ +\left(F\times Wk_{ppt}\right)+\left(G\times W_{ppt}\right)+H \end{array}$$

The postpartum weight (*W_pp_*) from measurement was already used as an independent variable in calculating the body circumference of postpartum women according to Equation (7). However, when a user wants to predict her 3D postpartum shape at other postpartum weeks using web app (Z-Size Ladies), postpartum weight is unknown. In the case of calculating the postpartum weight as a dependent variable, it would be complicated since it involved many factors such as BMI, pre-pregnancy weight, gestational weight gain, baby weight, and postpartum age. Moreover, data must be divided into four groups according to BMI types (underweight, normal weight, overweight and obese). Therefore, data from 83 postpartum women must also be divided into four groups, resulting in less than 30 postpartum women in each group. Data with n < 30 may not be sufficient for statistical analysis calculations [51]. 

Therefore, in the postpartum simulation application, the postpartum weight (*Wpp*) was predicted from the review articles. Theananansuk and Lertbunnaphong [52] concluded that the mean weight retention at the sixth week postpartum in Thai singleton pregnancy with normal pre-pregnancy BMI was 4.99 Kg. Cheng and Schmitt [53,54] discussed the postpartum weight retention in Asia and reviewed other articles showing that the postpartum weight retention at 0–24 weeks with inversion was approximately 7.4–2.5 Kg. Huang [8] studied 602 postpartum Taiwanese women and provided gestational weight gain (GWG), body weight retention and BMI at six months postpartum. Therefore, GWG from Huang [8] was compared to the values from IOM 2009 [46] to calculate GWG as shown in Table 5. The calculated weight retention results at six-month postpartum are shown in Table 6. Comparing results from six-month weight retention calculated for this study by applying the rule of three in arithmetic in comparison with the corresponding results from Huang [8] and IOM 2009 [46] have shown the weight retention. The results implied that females with extreme levels of BMI were slightly more vulnerable from higher weight retention than Taiwan females with corresponding BMI types, while the results were on the reverse for the case of normal weight and overweight.

An article by American Pregnancy Association (APA) [55] provided an average pregnancy weight gain distribution in a total of 30 pounds (13.63 Kg) as shown in Table 7.

From the above assumption, the weight retention at zero weeks (or delivery date) would be about half of the GWG (Average total weight gain from IOM), which were 7.7275, 6.8175, 4.545 and 3.5225 for pre-pregnancy BMI of underweight, normal weight, overweight and obese, respectively. Therefore, the weight retention from 0–24 weeks can be calculated by fitting graphs with different BMI types, as shown in Figure 4. The corresponding equations could be expressed as shown in Equations (8)–(11) with *X* referring to postpartum weeks while *Y_UW_*, *Y_NW_*, *Y_OW_*, *Y_OB_* are postpartum weight retention for the case of pre-pregnancy BMI type: underweight, normal weight, overweight and obese, respectively. In addition, postpartum weight was calculated as a summation of pre-pregnancy weight and weight retention.
(8)$$Y_{UW}=0.0064X^{2}-0.3268X+7.7275$$
(9)$$Y_{NW}=0.0068X^{2}-0.3452X+6.8175$$
(10)$$Y_{OW}=0.0052X^{2}-0.2662X+4.5450$$
(11)$$Y_{OB}=0.0057X^{2}-0.2915X+3.5225$$

### 3.5. Simulation of Postpartum Women in 3D

The 3D model of a postpartum woman was calculated in a similar way to the non-pregnant female [42], although with different parameters. For simulating a postpartum body shape, the morphing technique was applied, as in Equation (12), which was similar to Equation (2).
(12)$$P_{i}=(1-\overset{5}{\underset{i=1}{\sum }}K_{i})\times A_{i}+\left(K_{1}\times B_{i}\right)+\left(K_{2}\times C_{i}\right)+\left(K_{3}\times D_{i}\right)+\left(K_{4}\times E_{i}\right)+\left(K_{5}\times F_{i}\right)$$
where *A_i_–F_i_* are avatars with thin, big breast, big waist, big hip, tall and long legs, respectively. From the experiment, it was found that the *K*_1_–*K*_5_ values were also proportional to the BMI and also depended on the postpartum weeks as well. In our experiment, a second order polynomial fitting curve and parameters from the experiment were performed as shown in Equation (13a–c), which is similar to Equation (3a–d):(13a)$$K_{1}=K_{01}+Alp_{1},\;\;\;K_{2}=K_{02}+Alp_{2},\;K_{3}=K_{03}+Alp_{3},\;K_{4}=K_{04}+Alp_{4},\;K_{5}=K_{05}+Alp_{5}$$
(13b)$$\mathrm{Where}\;K_{01}=\frac{\left(Chest-57\right)}{\left(200-57\right)},\;K_{02}=\frac{\left(Waist-40\right)}{\left(160-40\right)},$$
(13c)$$K_{03}=\frac{\left(Hip-68\right)}{\left(180-68\right)},\;K_{04}=\frac{1.36\times \left(Height-165\right)}{\left(199-165\right)},\;K_{05}=\frac{\left(Inseam-78\right)}{\left(120-48\right)}\;$$

The values of Chest, Waist, Hip and Inseam are the values calculated from non-pregnant female body shape according to the article by Sinthanayothin, et al. [42]. The values could be expressed as linear equations shown in Equation (14a–d), which are similar to the ones shown in Equation (4a–d).
(14a)Chest=0.872260×Weight−0.437949×Height+110.131573 
(14b)Waist=0.931735×Weight−0.497702×Height+104.780946 
(14c)Hip=0.729978×Weight−0.152380×Height+78.526838 
(14d)Inseam=−0.059182×Weight+0.547734×Height−14.226815 

From the experiments of cross-sectioning and measuring the circumference of 3D simulation figures, *Alp*_1_–*Alp*_5_ were functions of the body mass index (BMI), which could be calculated as the following morph equations, which were quadratic functions shown in Equation (15a–f). They are similar to those shown in Equation (5a–e).
(15a)$$Alp_{1}=\left(Alp_{11}\times BMI\times BMI\right)+\left(Alp_{12}\times BMI\right)+Alp_{13}\;$$
(15b)$$Alp_{2}=\left(Alp_{21}\times BMI\times BMI\right)+\left(Alp_{22}\times BMI\right)+Alp_{23}\;$$
(15c)$$Alp_{3}=\left(Alp_{31}\times BMI\times BMI\right)+\left(Alp_{32}\times BMI\right)+Alp_{33}\;$$
(15d)$$Alp_{4}=\left(Alp_{41}\times BMI\times BMI\right)+\left(Alp_{42}\times BMI\right)+Alp_{43}\;$$
(15e)$$Alp_{5}=\left(Alp_{51}\times BMI\times BMI\right)+\left(Alp_{52}\times BMI\right)+Alp_{53}\;$$

*Alp_XY_* is a variable that depends on the postpartum weeks (*Wk_ppt_*), so it can be written as a quadratic function shown in Equation (16) and similar to Equation (6a).
(16)$$Alp_{XY}=\left(Alp_{XYC}\times Wkppt\times Wkppt\right)+\left(Alp_{XYB}\times Wkppt\right)+Alp_{XYA}$$
where $Alp_{XYA}$, $Alp_{XYB}$ and $Alp_{XYC\;}$ are the constants calculated by polynomial fitting shown in Table 8. These values were applied to the morphing equations to obtain 3D postpartum women model closed to the calculated statistical value, as shown in Equation (7).

## 4. Results

### 4.1. Statistical Correlation and 3D Modeling Simulation for Pregnant Female Body Shape

The results of the correlation analysis of pregnant women shape with independent variables using multiple linear regression of the 587 data collected from 98 pregnant women are shown in Table 9. Coefficient values were calculated according to the woman’s age, pre-pregnancy weight, height, gestational age, and weight gain during pregnancy, which were applied to the statistical calculation of pregnant female body shape in Equation (1).

Figure 5 shows the results of the pregnant women 3D simulations at various gestational ages for women with four types of pre-pregnancy BMI: Underweight (BMI < 18.5); Normal weight (18.5 ≤ BMI ≤ 24.9); Overweight (25.0 ≤ BMI ≤ 29.9); and Obese (BMI ≥ 30.0).

### 4.2. Statistical Correlation and 3D Modeling Simulation for Postpartum Female Body Shape

The results of the correlation analysis of postpartum women’s shape with independent variables using multiple linear regression of the 503 data collected from 83 postpartum females are shown in Table 10. Coefficient values were calculated according to the woman’s age, pre-pregnancy weight, height, gravida, baby weight, postpartum week, and postpartum weight, which were applied to the statistical calculation of postpartum female body shape in Equation (7).

Figure 6 shows the results of 3D simulation of postpartum women from the postpartum week of zero to 24 weeks with Height of 165 cm, Pre-pregnancy Weight of 60 Kg, Baby weight of 3 Kg.

### 4.3. The Accuracy Test on 3D Modeling of Pregnant Women

For accuracy test on pregnant simulation, the average and standard deviation of the measurements, Pearson coefficients and relative errors were calculated to measure the association and agreement between pairs of measurement methods. Furthermore, confidence interval plots were examined to assess and compare the results of the two methods.

The accuracy test on 3D modeling of pregnant women was divided into two parts:A.Comparison of body measurements in centimeters of chest, waist, hip, upper arm, thigh, and inseam between 3D models of pregnant women and the calculated statistical values for 30 datasets, with the mean age of 28.22 ± 4.69 years, the height of 159.13 ± 4.78 cm, the pre-pregnancy weight of 61.82 ± 18.19 Kg and gestational age of 12–36 weeks with the mean at 24 ± 9.9 weeks. Z = Z-Size Ladies statistical values, B = Manual cross-section on 3D model and measured body circumferences, Chest = chest, Waist = waist, Hip = hip, Thigh = thigh circumference, Upper Arm = upper arm circumference and Inseam = leg length, respectively. Where Avg is the mean, SD is the standard deviation, L CI and U CI are the lower and upper bounds of the 95% confidence interval, R Error is the relative error and Corr is the correlation, as shown in Table 11. Also 95% Confidence Interval (CI) plot for the mean measurements of chest, waist, hip, upper arm, thigh, and inseam of pregnant women between Z-Size Ladies statistical values and manual cross-section 3D values are shown in Figure 7.

B.Comparison of chest, waist, hip, and arm thigh in centimeters between the values calculated from the statistical data and the values obtained from 75 pregnant volunteers using a tape measurement at Maharaj Nakorn Chiang Mai Hospital, with the mean age of 29.72 ± 4.95, the height of 156.97 ± 5.32 cm, the pre-pregnancy weight of 55.37 ± 10.01 Kg and the mean gestational age at 6–39 weeks with the mean of 25.71 ± 9.9 weeks. When Z = Z-Size Ladies statistical values, M = Manual, Weight Z = estimated maternal weight with App, Weight M = actual mother’s weight, Chest = chest, Waist = waist, Hip = hip circumference and Upper Arm = upper arm circumference. Avg is the mean, SD is the standard deviation, L CI and U CI are the lower and upper bounds of the 95% confidence interval, R Error is the relative error and Corr is the correlation, as shown in Table 12. Also 95% Confidence Interval (CI) plot for the mean measurements of weight, chest, waist, hip, and upper arm of pregnant women between Z-Size Ladies statistical values and tape measurements are shown in Figure 8.

### 4.4. The Accuracy Test on 3D Modeling of Postpartum Women

Similarly to Section 4.3, in order to calculate the accuracy test for postpartum simulation, the average and standard deviation of the measurements, Pearson coefficients and relative errors were calculated to measure the association and agreement between pairs of measurement methods. Furthermore, confidence interval plots were examined to assess and compare the results of the two methods.

The accuracy test on 3D modeling of postpartum women was divided into two parts:A.Comparison of body proportion in centimeters of chest, waist, hip, upper arm, thigh, and inseam between 3D models of postpartum women and the calculated statistical values for 30 datasets, with the mean age of 27.4 ± 6.18 years, the height of 155.43 ± 5.67 cm, the pre-pregnancy weight of 57.13 ± 47.09 Kg and postpartum age of 0–24 weeks with the mean at 9 ± 12 weeks. Z = Z-Size Ladies statistical values, B = Manual cross-section on 3D model and measured body circumferences, Chest = chest, Waist = waist, Hip = hip, Thigh = thigh circumference, Upper Arm = upper arm circumference and Inseam = leg length. Avg is the mean, SD is the standard deviation, L CI and U CI are the lower and upper bounds of the 95% confidence interval, R Error is the relative error and Corr is the correlation, as shown in Table 13. Also 95% Confidence Interval (CI) plot for the mean measurements of chest, waist, hip, upper arm, thigh, and inseam of postpartum women between Z-Size Ladies statistical values and manual cross-section 3D values are shown in Figure 9.


B.Comparison of chest, waist, hip, and upper arm in centimeters between the values calculated from statistical data and the values obtained from 74 postpartum volunteers using a tape measurement at Maharaj Nakorn Chiang Mai Hospital, with the mean age of 29.90 ± 5.67, the height of 156.94 ± 6.07 cm, the pre-pregnancy weight of 57.26 ± 13.92 Kg and the mean postpartum age at 0–24 weeks with the mean of 7.02 ± 5.12 weeks. Z = Z-Size Ladies statistical values, M = Manual, Weight Z = estimated maternal weight with App, Weight M = actual mother’s weight, Chest = chest, Waist = waist, Hip = hip circumference and Upper Arm = upper arm circumference, respectively. Avg is the mean, SD is the standard deviation, L CI and U CI are the lower and upper bounds of the 95% confidence interval, R Error is the relative error and Corr is the correlation, as shown in Table 14. Also 95% Confidence Interval (CI) plot for the mean measurements of weight, chest, waist, hip, and upper arm of postpartum women between Z-Size Ladies statistical values and tape measurements are shown as in Figure 10.


### 4.5. Satisfaction Test of the Developed Tool That Helps Pregnant/Postpartum Women to Simulate Their 3D-Body Shapes, Based on Height, Weight, and Gestational Age (Web App Z-Size Ladies)

Survey results of the satisfaction test of the web app Z-Size Ladies by 149 pregnant and postpartum volunteers at Maharat Nakorn Chiang Mai Hospital are shown in Table 15, the highest score of each item was five.

## 5. Discussion

The results of the correlation analysis of pregnant/postpartum women’s body shape with independent variables: pre-pregnancy weight; height; gestational age/ postpartum duration; and weight gain are shown in Table 9 and Table 10, respectively. Errors that could occur during statistical modeling in this study could come from the multiple regression models in the independent variables [56] such as pre-pregnancy weight, height, age, gestation age, and body circumferences measurements data (chest, waist, hip). Our regression models assumed that those variables and data were obtained from measurement without errors. Moreover, errors could come from a small sample size, which might lead to insignificant results, whereas too large a sample size may increase the risk of harming volunteer subjects and might cause them discomfort [57].

Also, our developed technique can simulate 3D body shapes of women during pregnancy and postpartum in various gestational ages, BMI, and postpartum duration as shown in Figure 5 and Figure 6, respectively. The pregnancy simulation included various gestational ages starting from 12–40 weeks with four types of pre-pregnancy BMI: underweight; normal weight; overweight and obese; and the postpartum at 0–24 weeks. For pregnancy simulation, different types of pre-pregnancy BMI indicates differences in weight gain during pregnancy. Therefore, 3D simulation of pregnant women was simulated at various gestational ages for women with four types of pre-pregnancy BMI according to IOM 2009 [46]. 

Note that for Figure 7 and Figure 9, the y-axis is in cm and the x-axis represents body circumference measurements. For Figure 8 and Figure 10, the y-axis displays in kg for weight and cm for circumferences and, the units are in the square blanket under the x-axis after each parameter value. The values on the y-axis are quite wide in range due to the different sizes of the upper arm and hip being quite significant. 

Comparing results from the accuracy test on body measurements between the statistical values from this study (Z-Size Ladies) and the corresponding results taken from the manual measurement of the cross-section of the pregnant 3D model taken from Z-Size Ladies on 30 datasets are shown in Table 11. The accuracy is a measure of the degree of closeness of the measured or calculated value to its actual value. The percent relative errors are less than 3% with the maximum error being the upper arm (Relative error = 2.931%). However, the results show a strong correlation with the overlap plots of 95% confident interval between the results from Z-Size Ladies statistical values with the manual cross-section measurements of 3D models (Corr > 0.9). It implies that the results from the statistic values of Z-Size Ladies are comparable to the results from the manual cross-section measurements of the 3D model. 

The accuracy test on body measurements between Z-Size Ladies statistic values and the manual tape measurements from 75 volunteers with the gestational age of 6–39 weeks is shown in Table 12. Table 12 shows all relative errors less than 3%, and the maximum error is in the upper arm (Relative error = 2.685%). It indicates a high correlation and some overlapping plots of 95% confident interval between the results from Z-Size Ladies and the results from the tape measurement (Corr > 0.89) even though the correlation is slightly less than the cross-section measurement on the 3D models. This is due to the locations in the manual tape measurements and the locations of measurements by Z-Size Ladies causing the variations. The highest correlation with the least relative error is at the chest and waist measurements due to the relative ease of locating the level for girt measurements of the chest and waist. However, the measurement of the upper arm has the lowest correlation value (Cor = 0.893) due to the difficulties in locating the places for upper arm measurements under the armpits. 

Comparing results from the accuracy test on body measurements between the statistical values from Z-Size Ladies and the corresponding results taken from the manual measurement of the cross-section of postpartum 3D models taken from Z-Size Ladies for 30 datasets are shown in Table 13. The maximum relative error belongs to the position of the upper arm and thigh with 8.368% and 4.824%, respectively. Also, the lowest correlation and less overlapping plots of 95% confident interval are at the upper arm and thigh measurements due to the difficulty in locating the exact location for manual measurements of upper arms and thighs, which are near the armpits and crotch.

The accuracy test on body measurements between Z-Size Ladies statistic values and the manual tape measurements from 74 volunteers with postpartum age of 0–24 weeks is shown in Table 14. The maximum relative error belongs to the position of the upper arm (Relative error = 6.184%) and waist (Relative error = 3.934%). The 95% confident interval shows non overlapping in the position of the upper arm. The measurement of the upper arm also has the lowest correlation value (Cor = 0.736) as well.

Comparison the accuracy test between 3D modeling of pregnant and postpartum women, the 3D pregnant simulation shows a higher correlation between statistical values and 3D body measurements, less error, and the 95% confident interval plots show the intervals overlapped better than the postpartum shape simulation. Furthermore, Figure 9 and Figure 10 show the upper arm relative error rate is relatively high and the 95% confidence interval plots show the intervals are not overlapped. This indicates that the 3D simulation of pregnant women is more accurate than the simulation of postpartum women in this study. This may be due to postpartum women beginning to work after the delivery of their babies and, as such, the arm, thighs, and other muscles become distinctly different from the calculated values, leading to variations in the upper arm and other body measurements.

This study presents a 3D model shape simulation of pregnant and postpartum women. The data of woman’s anthropometric measurements in different gestational and postpartum stages were collected. Based on the work of our previous study [42], pregnancy data was included to generate models to predict the shape of women at specific pregnancy and postpartum periods, based on pre-pregnancy measurements. The work led to the creation of a web application (Z-Size Ladies) to display 3D pregnancy and postpartum models, allowing women to input their metrics and observe the simulation. The website was validated through a survey from the users and received positive satisfaction scores from pregnant and postpartum women as illustrated in Table 15.

The limitations of this study were the resource deficiencies and the small sample size. Our research was a long-term perspective study that collected data six months before and after pregnancy. It was time-consuming and demanding to the participants. Therefore, the drop-out rate was high. The study excluded all possible aspects, such as our volunteers were pregnant and postpartum women with single pregnancy, who may not have regular exercise and may not have disabilities. In addition, this study included only a sample of Asians and did not include any foreigners.

## 6. Conclusions

Our web app (Z-size Ladies) accurately predicts the body proportions of pregnant and postpartum women based on a woman’s age (years), pre-pregnancy weight (Kg), height (cm), gravida (number of pregnancies, the default is one), pregnancy/gestational week (weeks) and weight gain (program predicted automatically with adjustable personalized input from the user). The experiment results have shown that Z-Size Ladies could generate 3D models of pregnant participants, as well as postpartum participants, with high accuracy and could be considered as a lower-cost alternative method to the use of a full-body 3D scanner. However, more participants are needed to ensure continuity and high statistics for the study and to improve the accuracy of 3D models for pregnant and postpartum women. Better algorithms for 3D data reconstruction on the obscured sections, such as armpits and thighs, would be required for improving the accuracy of upper arm and thigh measurements.

## Figures and Tables

**Figure 1 sensors-22-02036-f001:**
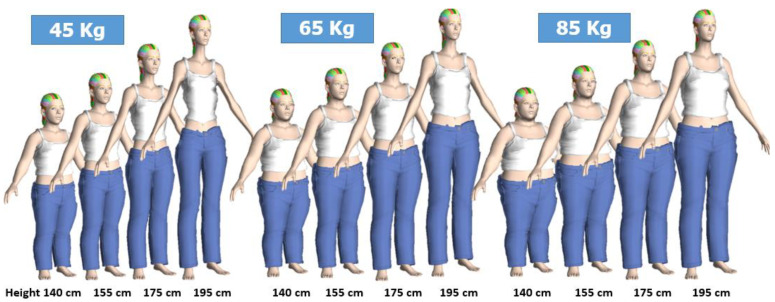
3D body simulation for non-pregnant females using Morphing Technique.

**Figure 2 sensors-22-02036-f002:**
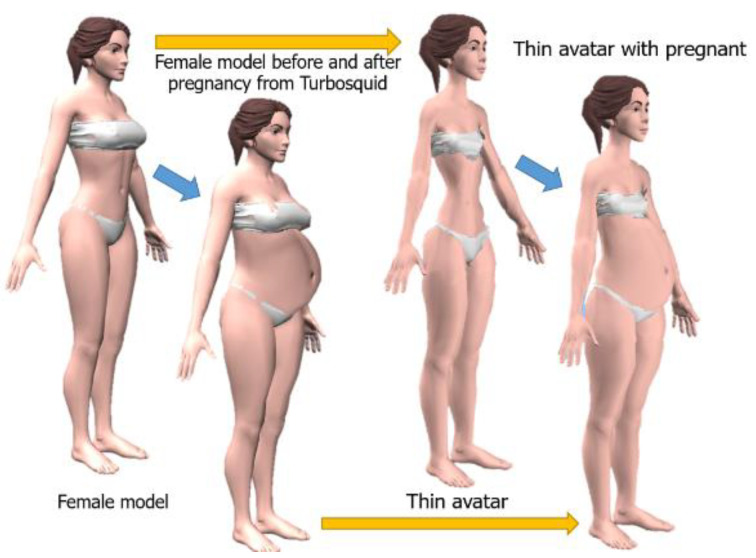
Creating a thin pregnancy avatar.

**Figure 3 sensors-22-02036-f003:**
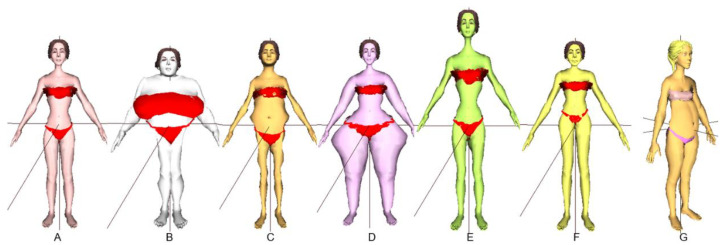
Seven avatars with accessories (eyes, eyes brows, hair, cloths): (**A**) thin, (**B**) big breast, (**C**) big waist, (**D**) big hip, (**E**) tall avatar, (**F**) long legs and (**G**) pregnant thin avatar. (Avatar (**G**)) has different view/pose as we would like to emphasize that this avatar has been added to this study while other avatars are from our previous study. Avatars (**A**–**F**) are shown in ‘front view’. However, if avatar (**G**) is shown in only ‘front view’, the shape changes from pregnancy would be difficult to notice).

**Figure 4 sensors-22-02036-f004:**
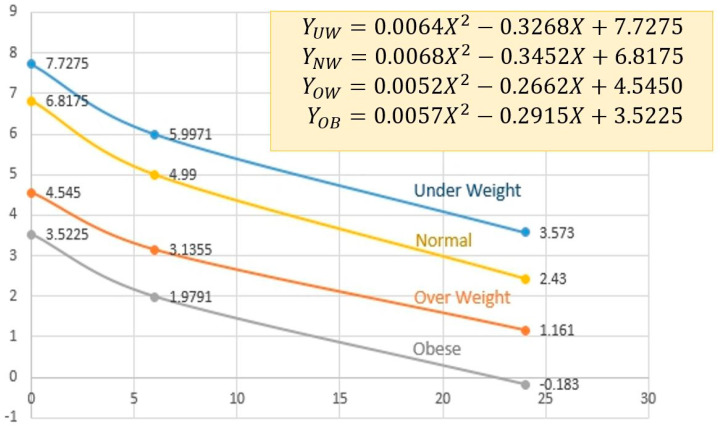
Weight retention estimation from 0 to 24 weeks postpartum, with pre-pregnancy BMI types.

**Figure 5 sensors-22-02036-f005:**
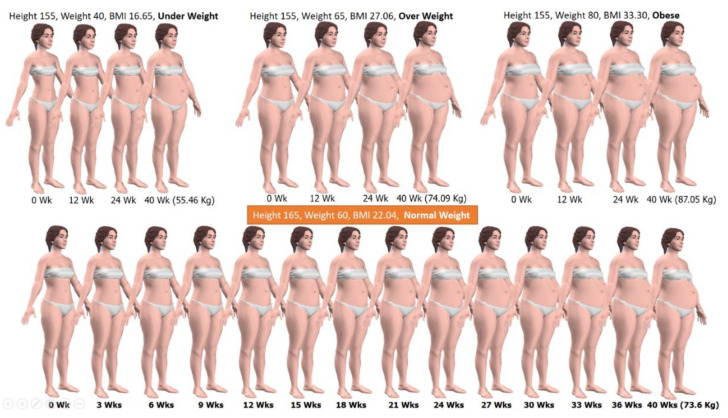
The simulated 3D models before pregnancy and during pregnancy 12, 24, and 40 weeks with four types of pre-pregnancy BMI: underweight, normal weight, overweight and obese.

**Figure 6 sensors-22-02036-f006:**
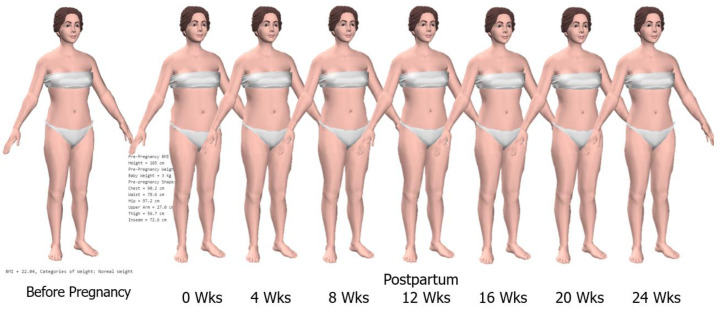
3D simulation models of postpartum women.

**Figure 7 sensors-22-02036-f007:**
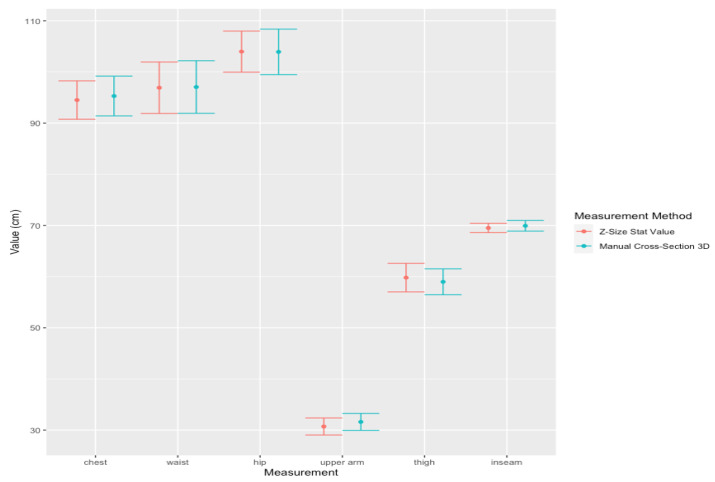
95% Confidence Interval (CI) plot for the mean measurements (30 data from pregnant) of chest, waist, hip, upper arm, thigh, and inseam between Z-Size Ladies statistical values and manual cross-section 3D values.

**Figure 8 sensors-22-02036-f008:**
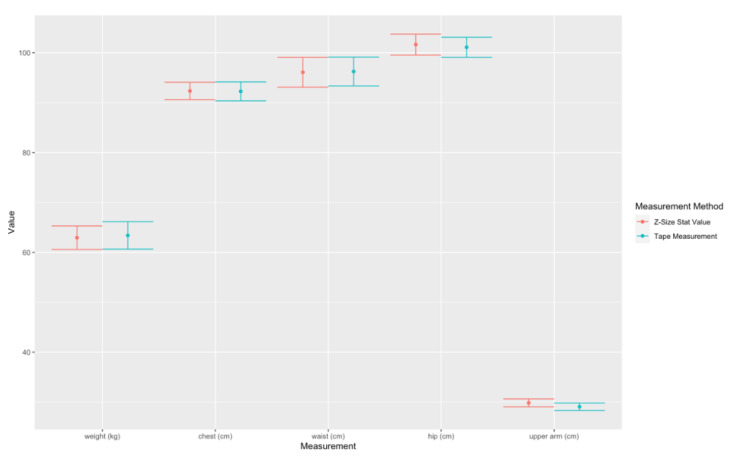
95% Confidence Interval (CI) plot for the mean measurements (from 75 pregnant volunteers) of weight, chest, waist, hip, and upper arm between Z-Size Ladies statistical values and tape measurements.

**Figure 9 sensors-22-02036-f009:**
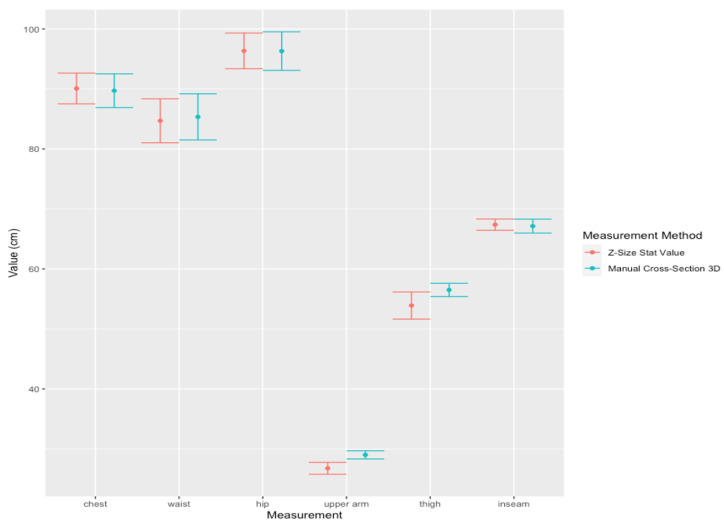
95% Confidence Interval (CI) plot for the mean measurements (30 data from postpartum) of chest, waist, hip, upper arm, thigh, and inseam between Z-Size Ladies statistical values and manual cross-section 3D values.

**Figure 10 sensors-22-02036-f010:**
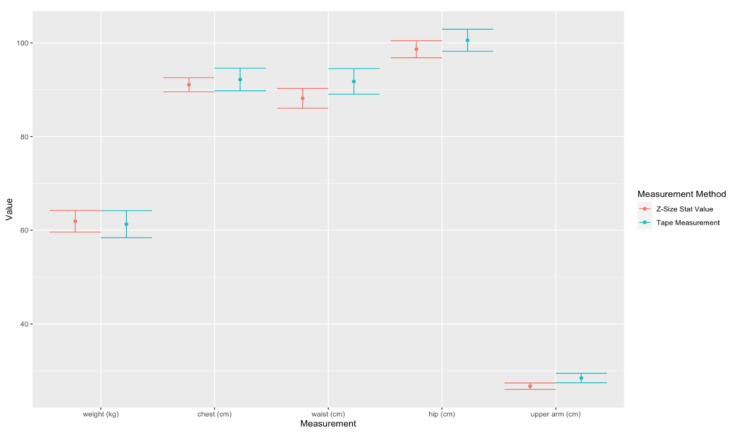
95% Confidence Interval (CI) plot for the mean measurements (74 postpartum volunteers) of weight, chest, waist, hip, and upper arm between Z-Size Ladies statistical values and tape measurements.

**Table 1 sensors-22-02036-t001:** Information the pregnant participants.

Information	Range	Average	SD
Age (*Y_w_*—Years)	18–43.5	29.64	5.24
Pre-Pregnancy Weight (*W_pp_*—Kg)	38–102	54.07	11.30
Height (*H_w_*—cm)	104–174	157.58	5.97
Gravida (*Gr*—child)	1–2	1.39	0.49
Pregnancy Week (*Wk_p_*—Weeks)	12–36	23.05	7.88
Weight Gain (*W_g_*—Kg)	−5–26	6.33	5.01

**Table 2 sensors-22-02036-t002:** Weight gained during pregnancy (kg) at each pregnancy stage based on pre-pregnancy BMI.

Pre-Pregnancy BMI Type	Pre-Pregnancy BMI (Kg/m^2^) (WHO)	Weight Gain (Kg)	Weight Gain per Week during the Quarter 2–3 (Kg/Wk)
Under weight	<18.5	12.73–18.18	0.45 (0.45–0.59)
Normal weight	18.5–24.9	11.36–15.91	0.45 (0.36–0.45)
Over weight	25.0–29.9	6.82–11.36	0.27 (0.23–0.32)
Obese	≥30.0	5.00–9.09	0.23 (0.18–0.27)

**Table 3 sensors-22-02036-t003:** Correlation coefficient $Alp_{XY}$ used in morphing equations.

$Alp_{XY}$	$Alp_{XYA}$	$Alp_{XYB}$	$Alp_{XYC}$
$Alp_{01}$	−6.956 × 10^−3^	4.582 × 10^−4^	−8.577 × 10^−6^
$Alp_{02}$	4.671 × 10^−1^	−2.857 × 10^−2^	5.314 × 10^−4^
$Alp_{03}$	−6.945	3.811 × 10^−1^	−6.870 × 10^−3^
$Alp_{11}$	−1.033 × 10^−4^	−2.141 × 10^−5^	6.106 × 10^−7^
$Alp_{12}$	5.440 × 10^−3^	9.985 × 10^−4^	−2.975 × 10^−5^
$Alp_{13}$	−1.019 × 10^−1^	−9.765 × 10^−3^	3.342 × 10^−4^
$Alp_{21}$	−3.492 × 10^−4^	9.483 × 10^−5^	−2.375 × 10^−6^
$Alp_{22}$	1.685 × 10^−2^	−7.334 × 10^−3^	1.539 × 10^−4^
$Alp_{23}$	−1.692 × 10^−1^	1.215 × 10^−1^	−2.524 × 10^−3^
$Alp_{31}$	−1.053 × 10^−5^	2.944 × 10^−5^	−3.744 × 10^−7^
$Alp_{32}$	−2.220 × 10^−3^	−2.386 × 10^−3^	4.237 × 10^−5^
$Alp_{33}$	1.524 × 10^−2^	4.118 × 10^−2^	−9.049 × 10^−4^
$Alp_{41}$	1.083 × 10^−4^	−2.414 × 10^−5^	6.383 × 10^−7^
$Alp_{42}$	−4.416 × 10^−3^	1.477 × 10^−3^	−3.730 × 10^−5^
$Alp_{43}$	8.753 × 10^−2^	−2.232 × 10^−2^	5.554 × 10^−4^
$Alp_{51}$	−1.808 × 10^−4^	9.690 × 10^−6^	6.453 × 10^−7^
$Alp_{52}$	1.370 × 10^−2^	−1.370 × 10^−3^	−1.147 × 10^−5^
$Alp_{53}$	−1.656 × 10^−1^	2.241 × 10^−2^	−1.634 × 10^−5^

**Table 4 sensors-22-02036-t004:** Information the postpartum participants.

Information	Range	Average	SD
Age (*Y_w_*—Years)	17.1–45.25	29.05	5.17
Pre-Pregnancy Weight (*W_pp_*—Kg)	38–95	56.35	12.10
Height (*H_w_*—cm)	142–173	156.85	5.99
Gravida (*Gr*—child)	1–2	1.41	0.49
Baby Weight (*W_b_*—Kg)	2.1–4.02	2.97	0.42
Postpartum Week (*Wk_ppt_*—Weeks)	0–24	11.37	7.94
Postpartum Weight (*W_ppt_*—Kg)	35.5–117	59.51	12.19

**Table 5 sensors-22-02036-t005:** Comparison of gestational weight gain (GWG) for different types of BMI, obtained from two sources [8,46].

Pre-Pregnancy BMI Type	Pre-Pregnancy BMI (Kg/m^2^) (WHO)	Weight Gain (Kg)	GWG (Average Total Weight Gain (Kg) from IOM)	GWG (Kg) from Huang et al., 2010
Under weight	<18.5	12.73–18.18	15.455	14.36
Normal weight	18.5–24.9	11.36–15.91	13.635	14.37
Overweight	25.0–29.9	6.82–11.36	9.09	13.07
Obese	≥30.0	5.00–9.09	7.045	11.15

**Table 6 sensors-22-02036-t006:** Comparison of weight retention at six-month postpartum for different types of BMI.

Pre-Pregnancy BMI Type	Pre-Pregnancy BMI (Kg/m^2^) (WHO)	Weight Gain (Kg) (IOM 2009)	Weight Retention at 6-mo Postpartum (Huang et al., 2010)	Weight Retention at 6-mo Postpartum (Apply in This Study)
Under weight	<18.5	12.73–18.18	3.32	3.573
Normal weight	18.5–24.9	11.36–15.91	2.57	2.430
Overweight	25.0–29.9	6.82–11.36	1.67	1.161
Obese	≥30.0	5.00–9.09	−0.29	−0.183

**Table 7 sensors-22-02036-t007:** Average pregnancy weight gain distribution in a total of 13.63 Kg (30 pounds) suggested by APA. About half belongs to Mom and the other half belongs to the baby.

Pregnancy Weight Gain Distribution	Weight (Kg)	Belongs to Mom or Baby
The weight of the baby by the end of pregnancy	3.4	Baby
The weight of the placenta	0.68	Baby
Attributed to increased fluid volume	1.82	Baby
Increased blood volume	1.82	Baby/Mom
The weight of the uterus	0.91	Mom
The weight of breast tissue	0.91	Mom
Maternal stores of fat, protein and other nutrients	3.18	Mom
The amniotic fluid	0.91	Mom

**Table 8 sensors-22-02036-t008:** $Alp_{XY}$ correlation coefficients used in morphing equations for 3D postpartum woman model.

$Alp_{XY}$	$Alp_{XYA}$	$Alp_{XYB}$	$Alp_{XYC}$
$Alp_{11}$	3.942 × 10^−4^	−6.943 × 10^−6^	1.720 × 10^−7^
$Alp_{12}$	−2.351 × 10^−2^	5.974 × 10^−4^	−1.915 × 10^−5^
$Alp_{13}$	2.716 × 10^−1^	−8.912 × 10^−3^	3.394 × 10^−4^
$Alp_{21}$	−1.201 × 10^−3^	6.315 × 10^−5^	−1.935 × 10^−6^
$Alp_{22}$	6.153 × 10^−2^	−2.718 × 10^−3^	8.334 × 10^−5^
$Alp_{23}$	−5.699 × 10^−1^	1.173 × 10^−2^	−5.636 × 10^−4^
$Alp_{31}$	1.668 × 10^−5^	−1.037 × 10^−5^	3.959 × 10^−7^
$Alp_{32}$	−4.785 × 10^−3^	6.078 × 10^−4^	−1.986 × 10^−5^
$Alp_{33}$	7.614 × 10^−2^	−7.619 × 10^−3^	1.400 × 10^−4^
$Alp_{41}$	−6.535 × 10^−5^	2.052 × 10^−5^	−9.603 × 10^−7^
$Alp_{42}$	3.961 × 10^−3^	−5.569 × 10^−4^	3.121 × 10^−5^
$Alp_{43}$	−2.846 × 10^−2^	1.740 × 10^−3^	−1.908 × 10^−4^
$Alp_{51}$	−3.400 × 10^−4^	−1.750 × 10^−5^	7.106 × 10^−7^
$Alp_{52}$	1.718 × 10^−2^	2.525 × 10^−4^	−1.226 × 10^−5^
$Alp_{53}$	−1.914 × 10^−1^	1.244 × 10^−2^	−4.478 × 10^−4^

**Table 9 sensors-22-02036-t009:** Coefficient values for the multiple linear regression of pregnant women.

Value	A	B	C	D	E	F	G
Chest	−0.016	0.674	−0.132	0.090	0.100	0.457	69.896
Waist	0.108	0.752	−0.203	1.315	0.591	0.840	59.394
Hip	−0.022	0.736	−0.012	−0.491	−0.074	0.970	58.071
Upper Arm	0.017	0.306	−0.075	0.004	0.016	0.272	21.423
Thigh	0.031	0.516	−0.026	−0.064	−0.066	0.668	29.115

**Table 10 sensors-22-02036-t010:** Coefficient values for the multiple linear regression of postpartum women.

Value	*A*	*B*	*C*	*D*	*E*	*F*	*G*	*H*
Chest	−0.036	0.033	0.013	0.689	−0.066	−0.066	0.550	53.038
Waist	0.041	−0.064	−0.213	2.880	−2.098	−0.396	0.870	75.729
Hip	0.058	−0.074	−0.180	−0.022	−0.387	−0.196	0.780	83.935
Upper Arm	0.002	−0.014	−0.164	0.984	−0.533	−0.013	0.297	35.833
Thigh	−0.046	0.162	−0.124	−1.189	1.112	−0.072	0.390	41.065

**Table 11 sensors-22-02036-t011:** Comparing the average results of chest, waist, hip, upper arm circumference, thigh, and inseam, with the standard deviation, the lower and upper bounds of the 95% confidence interval, the relative error, and the correlation between the Z-Size Ladies statistical value and the circumference values measured cross-sectionally on 3D modelling for the case of pregnant women.

	Chest		Waist		Hip		Upper Arm		Thigh		Inseam	
	Z	B	Z	B	Z	B	Z	B	Z	B	Z	B
Avg	94.51	95.30	96.92	97.06	103.98	103.93	30.70	31.60	59.80	58.98	69.51	69.93
SD	10.49	10.86	14.05	14.36	11.22	12.42	4.66	4.63	7.79	7.06	2.56	2.91
L CI	90.76	91.41	91.89	91.92	99.96	99.49	29.03	29.94	57.01	56.45	68.59	68.89
U CI	98.26	99.19	101.95	102.20	108.00	108.37	32.37	33.26	62.59	61.51	70.43	70.97
R Error	0.836%		0.144%		0.048%		2.931%		1.371%		0.604%	
Corr	0.989		0.962		0.961		0.968		0.902		0.960	

**Table 12 sensors-22-02036-t012:** Comparing the average weight measurement, chest, waist, hip, and upper arm circumference with the standard deviation, the lower and upper bounds of the 95% confidence interval, the relative error and the correlation between the Z-Size Ladies statistical value and the value measured by tape measurement on 75 pregnant volunteers.

	Weight		Chest		Waist		Hip		Upper Arm	
	Z	M	Z	M	Z	M	Z	M	Z	M
Avg	62.94	63.40	92.36	92.27	96.10	96.26	101.65	101.11	29.83	29.05
SD	10.41	12.11	7.71	8.40	13.23	12.77	9.30	8.89	3.51	3.35
L CI	60.58	60.66	90.62	90.37	93.11	93.37	99.55	99.10	29.04	28.29
U CI	65.30	66.14	94.10	94.14	99.09	99.15	103.75	103.12	30.62	29.81
R Error	0.725%		0.098%		0.166%		0.534%		2.685%	
Corr	0.960		0.940		0.963		0.917		0.893	

**Table 13 sensors-22-02036-t013:** Comparing the average results of chest, waist, hip, upper arm circumference, thigh, and inseam, with standard deviation and the correlation between the Z-Size Ladies statistical value and the circumference values measured cross-sectionally on 3D modeling for the case of postpartum women.

	Chest		Waist		Hip		Upper Arm		Thigh		Inseam	
	Z	B	Z	B	Z	B	Z	B	Z	B	Z	B
Avg	90.08	89.71	84.70	85.35	96.34	96.31	26.77	29.01	53.90	56.50	67.38	67.14
SD	7.17	7.85	10.23	10.75	8.31	8.98	2.77	1.89	6.32	3.09	2.62	3.24
L CI	87.51	86.90	81.04	81.50	93.37	93.10	25.78	28.33	51.64	55.39	66.44	65.98
U CI	92.65	92.52	88.36	89.20	99.31	99.52	27.76	29.69	56.16	57.61	68.32	68.30
R Error	0.411%		0.767%		0.031%		8.368%		4.824%		0.356%	
Corr	0.963		0.979		0.992		0.882		0.969		0.985	

**Table 14 sensors-22-02036-t014:** Comparing the average weight measurement, chest, waist, hip and upper arm circumference with the standard deviation and the correlation between the Z-Size Ladies statistical value and the value measured by tape measurement on 74 postpartum volunteers.

	**Weight**		**Chest**		**Waist**		**Hip**		**Upper Arm**	
	**Z**	**M**	**Z**	**M**	**Z**	**M**	**Z**	**M**	**Z**	**M**
Avg	61.92	61.29	91.06	92.18	88.16	91.77	98.63	100.56	26.70	28.46
SD	10.16	12.69	6.59	10.56	9.22	11.95	7.95	10.33	3.06	4.42
L CI	59.61	58.40	89.56	89.77	86.06	89.05	96.82	98.21	26.00	27.45
U CI	64.23	64.18	92.56	94.59	90.26	94.49	100.44	102.91	27.40	29.47
R Error	1.028%		1.215%		3.934%		1.919%		6.184%	
Corr	0.891		0.847		0.828		0.901		0.736	

**Table 15 sensors-22-02036-t015:** Satisfaction test for using Z-Size Ladies, surveyed from 149 pregnant and postpartum volunteers.

Information	Rating Stars
This application is interesting.	4.66
Working efficiency such as fast response.	4.44
Ease of use.	4.32
Layout, keypad size, icon placement on screen.	4.19
Would you recommend this app to other pregnant women?	4.51
Do you think you will use the app again during pregnancy or after delivery?	4.56
How many stars would you rate the average for this app?	4.50

## Data Availability

Not applicable.

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
