# Peer review of "Simulation of 3D Body Shapes for Pregnant and Postpartum Women"

_sensors, 2022, doi:10.3390/s22052036_

Round 1

Reviewer 1 Report

This is an interesting study, many thanks! I have only very few minor comments:

  • Please limit all coefficients in all tables etc. to 3 digits only.
  • Could you please add a paragraph to the discussion about potential limitations of your study?

Author Response

Response to Reviewer 1 Comments

Point 1: This is an interesting study, many thanks! I have only very few minor comments:

Response 1: Thank you very much for your encouragement and insights, which helped us to improve our manuscript. We followed your suggestions as listed below:

Point 2: Please limit all coefficients in all tables etc. to 3 digits only.

Response 2: We changed all coefficients in all tables to 3 digits.

Point 3: Could you please add a paragraph to the discussion about potential limitations of your study?

Response 3: We added a paragraph to the discussion about the limitations of our study as follows. “The limitations of this study were the resource deficiencies and a small sample size. Our research was a long-term perspective study that collected data six months before and after pregnancy. It was time-consuming and demanding to the participants. Therefore, the drop-out rate was high. The study excluded all possible aspects such as our volunteers were pregnant and postpartum women with single pregnancy who may not have regular exercise and may be disabilities. In addition, this study included only a sample of Asians and did not include any foreigners.”

Reviewer 2 Report

The paper considers the shape changes in women before, during, and after pregnancy. Authors develop techniques to accurately predict the body proportions of pregnant and postpartum women based on simple tape anthropometric measurements.

General remarks:

The paper seems to be written by at least two different researchers since the language and clarity of presentation are changing. Therefore, I would suggest, researchers, consolidate the paper and give the paper to a native speaker for editing.

The structure and importance of each section could be also edited for improving the paper. While methods and materials seem to be quite extensive and some of it could be moved to the results section; the results section and especially the discussion section are rather sparse and short.

Specific remarks:

Abstract:

“Our objectives are to develop a technique that can compute and visualize 3D body shapes of women during pregnancy and postpartum in various forms.” – what forms? Explain in manuscript

“Changes in body according to 587 datasets from 98 pregnant and 503 datasets from 83  postpartum women were collected for 6 months tracking and analyzed for model shapes’ calculation.” – What is meant by datasets? Explain in the manuscript.

“Our results can predict accurately the body proportions of pregnant and postpartum women.” – based on what input? Provide this info in manuscript

1. Introduction:

“There are several researches looking at correlations of body shapes and BMI for women.” – please provide references.

“Nevertheless, there is no physical simulation or any way to find the shape proportion of pregnant and postpartum women, which are considered to be a population group with high health risks and vulnerability.” – what risk? Provide references…

“It is not possible to make a direct data 60 collection using the 3D body scanner.” – why not? Explain in the manuscript. Provide also reference.

“However, Body Visualizer has been using “Civilian American and European Surface Anthropometry Resource” (CAESAR) as the basis for creating Body Visualizer, which based on the data from SizeUSA while the result of the simulation is just a clay figure.” – vague sentence. Please edit the manuscript with a native English speaker.

3. Methodology

Figure 2. – what is meant with “Thin avatar with pregnant”?

Figure 3. – why different view/pose for avatar G?

4.3. The satisfaction test of the web app Z-Size Ladies v.2

This is not a part of this study and the authors also do not provide any discussion based on this. I suggest the authors remove this from the paper.

5. Discussion

“On the other hand, the lowest correlation is at the thigh measurement (Corr = 0.90191) due to the difficulty to locate the point for manual measurements of thigh, which have to be near the crotch.” – do you think this is the only error? What about the errors that could occur during statistical modeling? How do you account for that? Consider this also in all further occurrences. Please discuss this in the discussion section and provide relevant data.

Author Response

Response to Reviewer 2 Comments

Point 1: The paper considers the shape changes in women before, during, and after pregnancy. Authors develop techniques to accurately predict the body proportions of pregnant and postpartum women based on simple tape anthropometric measurements.

Response 1: We thank the reviewer for the careful and insightful review of our manuscript.

General remarks:

Point 2: The paper seems to be written by at least two different researchers since the language and clarity of presentation are changing. Therefore, I would suggest, researchers, consolidate the paper and give the paper to a native speaker for editing.

Response 2: This article was written in collaboration of multiple authors. We believe the manuscript should represent the views of all authors, not only the part of their own contribution. Hence, the content consistency and language might show discontinuity. We have carefully consolidated the manuscript and sent it to the Professional Authorship Center, Thailand National Science and Technology Development Agency (NSTDA) for review and revision. Thanks for your advice. We will submit the manuscript to editing services (MDPI Author Services) for revisions by native English-speaker in minor revision after completing major revision correction.

Point 3: The structure and importance of each section could be also edited for improving the paper. While methods and materials seem to be quite extensive and some of it could be moved to the results section; the results section and especially the discussion section are rather sparse and short.

Response 3: We are grateful for your suggestion. The following parts were moved from the Method section to the Result section:

-           Table 2. Coefficient values for the multiple linear regression of pregnant women.

-           Figure 4: The simulated 3D model before pregnancy and during pregnancy 12, 24, and 40 weeks with four types of pre-pregnancy BMI: underweight, normal weight, overweight and obese respectively.

-           Table 6. Coefficient values for the multiple linear regression of postpartum women.

-           Figure6: 3D simulation models of postpartum women.

Specific remarks:

Point 4: Abstract section:

Response 4: Edited Abstract (193 words)

Abstract: Several studies have reported that pre-pregnant women's body mass index (BMI) affects women’s weight gain with complications during pregnancy and the postpartum weight retention. It is important to control the BMI before, during and after pregnancy. Our objectives are to develop a technique that can compute and visualize 3D body shapes of women during pregnancy and postpartum in various gestational ages, BMI, and postpartum durations. Body changes data from 98 pregnant and 83 postpartum women were collected, tracked for 6 months, and analyzed to create 3D model shapes. This study allows users to simulate their 3D body shapes in real-time and online, based on weight, height, and gestational age, using multiple linear regression and morphing techniques. To evaluate the results, precision tests were performed on simulated 3D pregnant and postpartum women’s shapes. Additionally, a satisfaction test on the application was conducted on new 149 mothers. The accuracy of the simulation was tested on 75 pregnant and 74 postpartum volunteers in terms of relationships between statistical calculation, simulated 3D models and actual tape measurement of chest, waist, hip, and inseam. Our results can predict accurately the body proportions of pregnant and postpartum women.

Comment: Abstract section:

Point 5: Abstract section:

Response 5: We thank the reviewer for these suggestions. The abstract is limited in word count, so we edited the abstract with a short explanation regarding reviewer question and additional explanation in the Discussion section.

Point 6: “Our objectives are to develop a technique that can compute and visualize 3D body shapes of women during pregnancy and postpartum in various forms.” – what forms? Explain in manuscript

Response 6: We edited the Abstract with a short explanation as “Our objectives are to develop a technique that can compute and visualize 3D body shapes of women during pregnancy and postpartum in various gestational ages, BMI, and postpartum durations.”

Further explanation is also added in the Discussion section

“Also, our developed technique can simulate 3D body shapes of women during pregnancy and postpartum in various gestational ages, BMI, and postpartum duration as shown in figure 5 and 6, respectively. The pregnancy simulation included various gestational ages starting from 12-40 weeks with four types of pre-pregnancy BMI: under-weight, normal weight, overweight and obese, and the postpartum at 0-24 weeks.”

Point 7: “Changes in body according to 587 datasets from 98 pregnant and 503 datasets from 83 postpartum women were collected for 6 months tracking and analyzed for model shapes’ calculation.” – What is meant by datasets? Explain in the manuscript.

Response 7: The sentence has been revised to make it simple in the abstract to “Body changes data from 98 pregnant and 83 postpartum women were collected, tracked for 6 months and analyzed to create 3D model shapes.”

More details of 587 data collected from 98 pregnant women were explained in Section 3.2 as follows. “The data from 98 pregnant women volunteers were collected and analyzed. Information of all pregnant volunteers is shown in Table 1. The data were 587 sets in total. Each data contained woman's age, pre-pregnancy weight, height, gestational age, weight gain during pregnancy, inseam (measure once at 12–16 week pregnancy), and body circumference measurements: chest, waist, hip, upper arm (left/right), and thigh (left/right). The data used for this study were from 94 women who were 12-week gestation; 98 of 16-week gestation; 91 of 20-week gestation; 82 of 24-week gestation; 79 of 28-week gestation; 78 of 32-week gestation and 65 of 36-week gestation, a total of 587 sets.”

More detail of 503 data collected from 83 postpartum women volunteers were explained in Section 3.4 as follows. “The data from 83 postpartum women volunteers were collected and analyzed. Information of all postpartum volunteers was shown in Table 5. The data were 503 sets in total. Each data contains woman's age, pre-pregnancy weight, height, gravida, baby weight, postpartum week, postpartum weight, inseam (measure once at 0 week of postpartum), and body circumference measurements: chest, waist, hip, upper arm (left/right), and thigh (left/right). The data used for this study were from 81 women of 0-week postpartum; 76 of 4-week postpartum; 73 of 8-week postpartum; 72 of 12-week postpartum; 72 of 16-week postpartum; 70 of 20-week postpartum and 59 of 24-week postpartum, a total of 503 sets.”

Point 8: “Our results can predict accurately the body proportions of pregnant and postpartum women.” – based on what input? Provide this info in manuscript

Response 8: We add this paragraph to the Conclusion section. “Our web app (Z-Size Ladies)  accurately predicts the body proportions of pregnant and postpartum women based on woman’s age (years), pre-pregnancy weight (Kg), height (cm), gravida (number of pregnancies, the default is 1), pregnancy/ gestational week (weeks) and weight gain (program predicted automatically with adjustable personalized input from the user).”

  1. Introduction:

Point 9: “There are several researches looking at correlations of body shapes and BMI for women.” – please provide references.

Response 9: Thank you for pointing this out. We added three references as follows:

There are some research reports looking at correlations between body shapes and BMI for women [14, 15, 16].

References:

[14] Simona F.P., Elisabeta R.L., Cristian R. M.  Relation Between Body Shape And Body Mass Index. Procedia - Social and Behavioral Sciences. Volume 197, 25 July 2015, Pages 1458-1463.  DOI: 10.1016/j.sbspro.2015.07.095

[15] Kang N.E., Kim S.J., Oh Y.S., Jang S. The effects of body mass index and body shape perceptions of South Korean adults on weight control behaviors; Correlation with quality of sleep and residence of place. Nutr Res Pract. 2020 Apr; 14(2): 160–166. DOI:10.4162/nrp.2020.14.2.160

[16] Wells J.C., Treleaven P., Cole T.J. BMI compared with 3-dimensional body shape: the UK National Sizing Survey. The American Journal of Clinical Nutrition, Volume 85, Issue 2, February 2007, Pages 419–425. DOI: 10.1093/ajcn/85.2.419

Point 10: “Nevertheless, there is no physical simulation or any way to find the shape proportion of pregnant and postpartum women, which are considered to be a population group with high health risks and vulnerability.” – what risk? Provide references…

Response 10: Thank you, we have removed “high health risks”. The sentence was revised and an additional explanation with a reference was added.

“Nevertheless, there is no physical simulation for the shape proportion of pregnant and postpartum woman subjects as they are a population group with vulnerability to use the 3D body scanner for their data collection. The word “vulnerable” is in the context of human research protections. Pregnant women are considered vulnerable because of the involvement of the fetus that may be affected by the research and the fetus cannot give consent [17].”

Reference:

[17] Pregnant Women, Fetuses and Neonates as Vulnerable Population. Human Research Protection Program, 2009. Available online at https://cphs.berkeley.edu/policies_procedures/sc501.pdf

Point 11: “It is not possible to make a direct data collection using the 3D body scanner.” – why not? Explain in the manuscript. Provide also reference.

Response 11: Thank you for your question. We have clarified and rewritten the statement as follows:

“It was not possible to collect the body shape data using the 3D body scanner in this study. This is because most pregnant women are concerned about the safety of 3D body scanners and have questions about the scanner potential consequences at all stages of pregnancy [18]. It is difficult to obtain and approve applications for research projects involving human subjects from the IRB (Institutional Review Board), and especially when asking for consent forms from research subjects [17]. We also needed pregnant and postpartum women to help us for measurement their body circumferences at home every four weeks during pregnancy and postpartum.”

References:

[18] Pregnancy and Security Screening. Frequently Asked Questions in HPS  Specialists in Radiation Protection, 2016. Available online at https://hps.org/publicinformation/ate/faqs/pregnancyandsecurityscreening.html

Point 12: “However, Body Visualizer has been using “Civilian American and European Surface Anthropometry Resource” (CAESAR) as the basis for creating Body Visualizer, which based on the data from SizeUSA while the result of the simulation is just a clay figure.” – vague sentence. Please edit the manuscript with a native English speaker.

Response 12: Thank you for pointing this out to us. We have revised it as follows:

“Body Visualizer has used the SizeUSA dataset, American and European Surface Anthropometry Resource (CAESAR) as the basis for creating 3D body shapes visualization [28, 29]. It is a visualization tool for a parametric 3D body model that provides accurate metrically anthropomorphic measurements based on laser scans of thousands of people from different ethnicities. However, it is still lacking the 3D body models for pregnant and postpartum women, especially for Asian women. Therefore, our study focused on simulating 3D shapes of Thai pregnant and postpartum women.”

References

[28] Loper, M., Mahmood, N., Romero, J., Pons-Moll, G., & Black, M. J. (2015). SMPL: A skinned multi-person linear model. ACM transactions on graphics (TOG), 34(6), 1-16.

[29] Osman, A. A., Bolkart, T., & Black, M. J. (2020). Star: Sparse trained articulated human body regressor. In Computer Vision–ECCV 2020: 16th European Conference, Glasgow, UK, August 23–28, 2020, Proceedings, Part VI 16 (pp. 598-613). Springer International Publishing

  1. Methodology

Point 13: Figure 2. – what is meant with “Thin avatar with pregnant”?

Response 13: Thank you for the question. We have clarified and added the statement:

“Pregnant thin avatar refers to the pregnant women's simulation based on our previous studies [42]. Pregnant woman simulation is a combination of non-pregnant woman shape and pregnant shape. Because the non-pregnant female simulation is a combination of a thin avatar (Figure 3(A)) and other avatar shapes (including Figure 3 (B) big breast, (C) big waist, (D) big hip, (E) tall avatar, (F) long legs). So a pregnant thin avatar (Figure 2 or Figure 3 (G)) is added to simulate a pregnant woman.”

[42] Sinthanayothin C, Bholsithi W, Gansawat D, et al. Simulation of three-dimensional female body shapes with proportional representation for various weights and heights. SIMULATION. 2020;96(11):851-866. DOI:10.1177/0037549720944466

Point 14: Figure 3. – why different view/pose for avatar G?

Response 14: Avatar G has different view/pose in Figure 3 because we would like to emphasize that this avatar has been added to this study while other avatars are from our previous study. Avatars (A)-(F) are shown in ‘front view’. However, if avatar G is shown in only ‘front view’, the shape changes from pregnancy would be difficult to notice.

Point 15:  4.3. The satisfaction test of the web app Z-Size Ladies v.2

This is not a part of this study and the authors also do not provide any discussion based on this. I suggest the authors remove this from the paper.

Response 15: Thank you for the suggestion. Although the satisfaction test of the web app Z-Size Ladies was not directly part of this study, this manuscript presents a tool that helps pregnant women to simulate their 3D body shapes based on their personalized height, weight, and gestational age. The validation of our developed application was carried out by means of a survey received from users. We have not removed this part from the revised manuscript but provided more information in the discussion as follows.

“This study presents a 3D model shape simulation of pregnant and postpartum women. The data of woman's anthropometric measurements in different gestational and postpartum stages were collected. Based on the work of our previous study [42], pregnancy data was included to generate models to predict the shape of women at specific pregnancy and postpartum periods, based on pre-pregnancy measurements. The work led to the creation of a web application (Z-Size Ladies) to display 3D pregnancy and postpartum models, allowing women to input their metrics and observe the simulation. The website was validated through a survey from the users and received positive satisfaction scores from pregnant and postpartum women as illustrated in Table 15.”

[42] Sinthanayothin C, Bholsithi W, Gansawat D, et al. Simulation of three-dimensional female body shapes with proportional representation for various weights and heights. SIMULATION. 2020;96(11):851-866. DOI:10.1177/0037549720944466

  1. Discussion

Point 16:  “On the other hand, the lowest correlation is at the thigh measurement (Corr = 0.90191) due to the difficulty to locate the point for manual measurements of thigh, which have to be near the crotch.” – do you think this is the only error? What about the errors that could occur during statistical modeling? How do you account for that? Consider this also in all further occurrences. Please discuss this in the discussion section and provide relevant data.

Response 16: We fully agree with the reviewer. So, we added this paragraph and provided two more references to the discussion.

“Errors that could occur during statistical modeling in this study could come from the multiple regression models in the independent variables [56] such as pre-pregnancy weight, height, age, gestation age, and body circumferences measurements data (chest, waist, hip). Our regression models assumed that those variables and data were obtained from measurement without errors. Moreover, errors could come from a small sample size, which might lead to insignificant results, whereas a too large sample size may increase the risk of harming volunteer subjects and might cause them discomfort [57].”

References

[56] Errors-in-variables models. Wikipedia, 2022. Available online at https://en.wikipedia.org/wiki/Errors-in-variables_models

[57] Biau D. J., Kernéis S., and Porcher R. Statistics in Brief: The Importance of Sample Size in the Planning and Interpretation of Medical Research. Clin Orthop Relat Res. 2008 Sep; 466(9): 2282–2288. doi: 10.1007/s11999-008-0346-9

We have carefully consolidated the manuscript and sent it to the Professional Authorship Center, Thailand National Science and Technology Development Agency (NSTDA) for review and revision.

Reviewer 3 Report

Summary
=======

The authors present their work on simulating 3D shape of pregnant and post-partum women. The authors collected data of women's anthropometric measurements in different gestational stages and post-partum. Based on the work of a previous study, they include the pregnancy data to learn a model to predict a woman's body shape at specific pregnancy and post-partum stages, based on her pre-pregnancy measurements. The authors have created a web-app to display their model, allowing women to input their metrics and observe a simulation. The website was positively received by pregnant women. The simulation looks interesting.

Issues
======

1. The motivation of the work is not solid.

- The authors make claims that are not backed by citations, e.g. "Currently, obesity is increasing rapidly causing health issues, especially in pregnant women.", "If a woman in the post-partum period is unable to regulate her weight to her pre-pregnant weight within six months, she will increase her BMI until becoming obese." In their introduction they say this simulation model is necessary to "raise awareness among women as a measure to prevent being overweight", however, this seems to be the opinion of the authors because they do not provide previous research suggesting this need nor do the authors carry out a study to demonstrate this. 
- The authors write: "there is no physical simulation or any way to find the shape proportion of pregnant and postpartum women", of course, there are ways of obtaining this proportion, the most basic being measuring the people with a measuring tape.
- As a motivation of what differentiates their work from others (e.g. [7,8]), the authors claim that "our 3D simulation has clothing, face, hair and can add texture." It is unclear how are this features relevant for a body shape simulation.
- "The simulation of the body shape can raise an awareness of women and encourage them to prevent overweight behaviours before and throughout the pregnancy to maintain good health in the long run." This is a major claim, however, the authors do not present any evidence on this. 
- In their conclusion, they write for the first time that their model is meant as an alternative to body scanning. This is perhaps a better motivation, however, it requires to be backed by literature. 

2. There are missing citations and incorrect citations:

- "Then data were analysed and processed in simulation of 3D modelling of pregnant and postpartum women based on data of normal female shapes from SizeThailand." There is no reference to SizeThailand.
- The name "Z-Size Ladies v.1" is mentioned often but no citation is provided.
- Their explanation of the work of Michael J. Black in multiple citations [5-11] is wrong:
"However, Body Visualizer has been using “Civilian American and European Surface Anthropometry Resource” (CAESAR) as the basis for creating Body Visualizer, which based on the data from SizeUSA while the result of the simulation is just a clay figure." Just a clay figure? I suggest the authors to look at the following papers:

Loper, M., Mahmood, N., Romero, J., Pons-Moll, G., & Black, M. J. (2015). SMPL: A skinned multi-person linear model. ACM transactions on graphics (TOG)34(6), 1-16.

Osman, A. A., Bolkart, T., & Black, M. J. (2020). Star: Sparse trained articulated human body regressor. In Computer Vision–ECCV 2020: 16th European Conference, Glasgow, UK, August 23–28, 2020, Proceedings, Part VI 16 (pp. 598-613). Springer International Publishing

The body visualizer is just a visualization tool for a parametric 3D body model that provides metrically accurate anthropomorphic measurements based on laser scans of thousands of people from different ethnicities.

- There are relevant studies on prediction of 3D body shape during pregnancy. I suggest the authors also have a look at this paper for example:

Balasubramanian, M., & Robinette, K. (2020). Longitudinal anthropometric changes of pregnant women: dynamics and prediction. International Journal of Fashion Design, Technology and Education13(3), 231-237.

3. Unclear methodology

-"3D body shape simulation of normal women, pregnant and postpartum women shape with body proportions are predicted in real-time and online from weight, height and gestational age." What do you mean by 'predicted in real-time'?
- The authors say they collected anthropomorphic data in a 6 months time span, however it is not clear how or who collected the data (was there a specialist measuring the women, did they do self-measurements, etc.) Were these repeated measurements? Were the same participants measured in time? 
- "All analysed data were 587 datasets, including 94 women who were 12-week gestation; 98 of 16-week gestation; 91 of 20-week gestation; 82 of 24-week gestation; 79 of 28-week gestation; 78 of 32-week gestation and 65 of 36-week gestation" This information is hard to understand. 
- In order to create the 'pregnant body shape' the authors utilize a combination of the following 'bodies': "Details of creating a 3D normal female shape by combining thin, big breast, big waist, 210 big hip, tall and long legs avatars". It is unclear why this choice was made, or what supports making such a mix of 'attributes'/'3D body shapes'.
- On page 6, the authors write: "From the experiment, it was found that the K0 - K5 values were also proportional to the body mass index (BMI) and also depended on the 217 gestational age as well." Here, for the first time, a 'experiment' is mentioned but there is no explanation of it in the paper.
- Multiple equations contain constant values (e.g. equation 3), however, the rationale behind these constants is not explained in the paper.
- Authors make a differentiation in four categories of BMI:
"The 3D simulation of pregnant women can simulate at various gestational ages for women with four types of pre-pregnancy BMI: Underweight (BMI <18.5), Normal weight 256 (18.5 ≤ BMI ≤ 24.9), Overweight (25.0 ≤ BMI ≤ 29.9) and Obese (BMI ≥ 30.0)" 
However, to my understanding, there is no factor in the equations to account for these categories. Why is this differentiation being made?
- "The data of 83 postpartum women volunteers were collected and analysed". Are these some of the women who were part of the pregnancy group? 
- "The postpartum weight Wppt was predicted from the review articles". Why not from observations in your post-pregnancy group?
- The accuracy is reported by means of correlation coefficients, it would be in my opinion clearer to report also absolute errors, and in general a more thorough analysis would be helpful.
- The 'satisfaction' test is fine, however, at the beginning of the paper, the authors claim that their simulation will: (1) raise awareness in women, (2) encourage women to prevent overweight before and during pregnancy, (3) maintain long term health. None of these outcomes have been measured, thus, making it not possible to evaluate the actual success of the study (if these points are really the goal of the study).

5. Wording
The term "normal women" to describe women who are not pregnant or in the "post-partum" period is problematic.
The authors use the word "datasets" throughout their manuscript, in a way that I assume means just data or data points. e.g. "They used datasets from 25 pregnant Caucasian ladies". This should be revisited and corrected.
- In equation 6, gestational age is represented by the variable 'NV' but earlier in the paper (e.g. equation 1) it is represented by the variable Wkp.
- A general major issue is the lack of clarity in the general writing, structure and flow of the paper. It is difficult to fully follow the logic of the authors. 

Author Response

Response to Reviewer 3 Comments

Point 1: Summary

=======

The authors present their work on simulating 3D shape of pregnant and post-partum women. The authors collected data of women's anthropometric measurements in different gestational stages and post-partum. Based on the work of a previous study, they include the pregnancy data to learn a model to predict a woman's body shape at specific pregnancy and post-partum stages, based on her pre-pregnancy measurements. The authors have created a web-app to display their model, allowing women to input their metrics and observe a simulation. The website was positively received by pregnant women. The simulation looks interesting.

Response 1: We are thankful to the reviewer for encouragement and insightful comments to improve our manuscript.

Issues
======

Point 2:    1. The motivation of the work is not solid.

- The authors make claims that are not backed by citations, e.g.

"Currently, obesity is increasing rapidly causing health issues, especially in pregnant women.",

"If a woman in the post-partum period is unable to regulate her weight to her pre-pregnant weight within six months, she will increase her BMI until becoming obese."

Response 2: We thank the reviewer for this viewpoint. The sentences that explain our motivation are updated with the citations as follows:

 “Obesity during pregnancy is a serious health problem for women. Worldwide, obstetricians and midwives have confronted increasing obesity among pregnant women [1, 2].”

[1] Poston, L., Harthoorn, L., van der Beek, E. et al. Obesity in Pregnancy: Implications for the Mother and Lifelong Health of the Child. A Consensus Statement. Pediatr Res 69, 175–180 (2011). DOI: 10.1203/PDR.0b013e3182055ede

[2] Chen C, Xu X, Yan Y. Estimated global overweight and obesity burden in pregnant women based on panel data model. PLoS One. 2018 Aug 9;13(8):e0202183. DOI: 10.1371/journal.pone.0202183.

“It was reported that if a woman in the postpartum period was unable to regulate her weight to her pre-pregnant weight within six months, postpartum weight retention could predict future weight gain and long-term obesity [10]. Another study suggested that the BMI of a woman of more than six months postpartum would indicate the retaining extra body fluids produced during pregnancy, as well as extra fat during the first six months postpartum [11].”

[10] van der Pligt P, Willcox J, Hesketh KD, Ball K, Wilkinson S, Crawford D, Campbell K. Systematic review of lifestyle interventions to limit postpartum weight retention: implications for future opportunities to prevent maternal overweight and obesity following childbirth. Obes Rev. 2013 Oct;14(10):792-805. DOI: 10.1111/obr.12053.

[11] Arizona WIC (Women Infant and Children) Nutrition Care Guidelines: Breastfeeding and Postpartum Women, 2015. Available online at: https://azdhs.gov/documents/prevention/azwic/face-to-face/2015/jan/6-Breastfeeding-and-Postpartum-Women.pdf

Point 3: In their introduction they say this simulation model is necessary to

"raise awareness among women as a measure to prevent being overweight", however, this seems to be the opinion of the authors because they do not provide previous research suggesting this need nor do the authors carry out a study to demonstrate this.

Response 3: We thank the reviewer for this comment. We have updated the introduction with clearer statements with references as follows:

“Sui Z et al. reported a statistically significant indication that women with a high degree of body image dissatisfaction were more likely to have higher gestational weight gain [12]. Hill B et al. also reported that timing of pregnancy and body attitudes could predict gestational weight gain (GWG). The findings suggested that lower attractiveness in early-to-middle pregnancy was associated with higher GWG [13]. Therefore, in this work, the 3D body shape simulation of women before, during, and after pregnancy has been developed with an expectation that women will not misestimate their BMI to prevent being overweight during pregnancy and the postpartum period. 3D body simulation may encourage women to prevent overweight behaviours before and throughout pregnancy to maintain good health in the long run."

References:

[12] Sui Z., Turnbull D., Dodd J. Effect of body image on gestational weight gain in overweight and obese women. Women and Birth. Volume 26, Issue 4, December 2013, Pages 267-272. DOI: 10.1016/j.wombi.2013.07.001

[13] Hill B, Skouteris H, McCabe M, Fuller-Tyszkiewicz M. Body image and gestational weight gain: a prospective study. J Midwifery Womens Health. 2013 Mar-Apr;58(2):189-94. doi: 10.1111/j.1542-2011.2012.00227

Point 4: The authors write:

"There is no physical simulation or any way to find the shape proportion of pregnant and postpartum women", of course, there are ways of obtaining this proportion, the most basic being measuring the people with a measuring tape.

Response 4: We have revised the sentence and given an explanation with references.

Nevertheless, there is no physical simulation for the shape proportion of pregnant and postpartum woman subjects as they are a population group with vulnerability to use the 3D body scanner for data collection. The word “vulnerable” is in the context of human research protections. Pregnant women are considered vulnerable because of the involvement of the fetus that may be affected by the research and the fetus cannot give consent [17].

“It was not possible to collect the body shape data using the 3D body scanner in the study. This is because most pregnant women are concerned about the safety of 3D body scanners and have questions about the scanner potential consequences at all stages of pregnancy [18]. It is difficult to obtain an approval for applications for research projects involving human subjects from the Institutional Review Board (IRB), and especially when asking for consent forms for research subjects [17]. We also needed pregnant and postpartum women to measure their own body circumferences at home every four weeks during pregnancy and postpartum.

References:

[17] Pregnant Women, Fetuses and Neonates as Vulnerable Population. Human Research Protection Program, 2009. Available online at https://cphs.berkeley.edu/policies_procedures/sc501.pdf

[18] Pregnancy and Security Screening. Frequently Asked Questions in HPS  Specialists in Radiation Protection, 2016. Available online at https://hps.org/publicinformation/ate/faqs/pregnancyandsecurityscreening.html

Point 5: As a motivation of what differentiates their work from others (e.g. [7,8]), the authors claim that "our 3D simulation has clothing, face, hair and can add texture." It is unclear how are these features relevant for a body shape simulation.

Response 5: Thank you for pointing this out to us. We have revised the paragraph to be:

“Body Visualizer has used the dataset based on SizeUSA, American and European Surface Anthropometry Resource (CAESAR) as the basis for creating 3D body shapes visualization [28, 29]. It is a visualization tool for a parametric 3D body model that provides metrically accurate anthropomorphic measurements based on laser scans of thousands of people from different ethnicities. However, it is still lacking the 3D body models for pregnant and postpartum women especially for Asian women. Therefore, in this study we focused on simulating the 3D shape of pregnant and postpartum women for Thais.”

[28] Loper, M., Mahmood, N., Romero, J., Pons-Moll, G., & Black, M. J. (2015). SMPL: A skinned multi-person linear model. ACM transactions on graphics (TOG), 34(6), 1-16.

[29] Osman, A. A., Bolkart, T., & Black, M. J. (2020). Star: Sparse trained articulated human body regressor. In Computer Vision–ECCV 2020: 16th European Conference, Glasgow, UK, August 23–28, 2020, Proceedings, Part VI 16 (pp. 598-613). Springer International Publishing

Point 6: "The simulation of the body shape can raise an awareness of women and encourage them to prevent overweight behaviours before and throughout the pregnancy to maintain good health in the long run." This is a major claim, however, the authors do not present any evidence on this.

In their conclusion, they write for the first time that their model is meant as an alternative to body scanning. This is perhaps a better motivation, however, it requires to be backed by literature.

Response 6: Thank you for suggesting a better motivation. We have revised the sentence as follows:

“The simulation of the body shape from body measurements can be considered as a low-cost alternative to full-body 3D scanning [32].”

Reference:

[32] Gallucci A., Znamenskiy D., and Petkovic M. Prediction of 3D Body Parts from Face Shape and Anthropometric Measurements. Journal of Image and Graphics, Vol. 8, No. 3, September 2020. DOI: 10.18178/joig.8.3.67-74

Point 7: 2. There are missing citations and incorrect citations:

"Then data were analysed and processed in simulation of 3D modelling of pregnant and postpartum women based on data of normal female shapes from SizeThailand." There is no reference to SizeThailand.

Response 7: Thank you for your advice. We have added references to SizeThailand as follows:

Then data were analyzed and processed to create simulation of 3D modeling of pregnant and postpartum women based on data of non-pregnant female shapes from SizeThailand [19, 20].

References

[19] Wells, J., Treleaven, P. & Charoensiriwath, S. Body shape by 3-D photonic scanning in Thai and UK adults: comparison of national sizing surveys. Int J Obes 36, 148–154 (2012). DOI: 10.1038/ijo.2011.51

[20] Wells JC, Charoensiriwath S, Treleaven P. Reproduction, aging, and body shape by three-dimensional photonic scanning in Thai men and women. Am J Hum Biol. 2011 May-Jun;23(3):291-8. DOI: 10.1002/ajhb.21151.

Point 8: The name "Z-Size Ladies v.1" is mentioned often but no citation is provided.

Response 8: We updated the manuscript by removing v.1, v.2 and left with only Z-Size Ladies and referred to reference [31].

Reference

[31] Z-Size Ladies: Web Application for BMI Timeline & Program assesses ladies’ sizes and facial images. 2021. Available at: https://zsize.openservice.in.th/

Point 9: Their explanation of the work of Michael J. Black in multiple citations [5-11] is wrong:

"However, Body Visualizer has been using “Civilian American and European Surface Anthropometry Resource” (CAESAR) as the basis for creating Body Visualizer, which based on the data from SizeUSA while the result of the simulation is just a clay figure." Just a clay figure? I suggest the authors to look at the following papers:

Loper, M., Mahmood, N., Romero, J., Pons-Moll, G., & Black, M. J. (2015). SMPL: A skinned multi-person linear model. ACM transactions on graphics (TOG), 34(6), 1-16.

Osman, A. A., Bolkart, T., & Black, M. J. (2020). Star: Sparse trained articulated human body regressor. In Computer Vision–ECCV 2020: 16th European Conference, Glasgow, UK, August 23–28, 2020, Proceedings, Part VI 16 (pp. 598-613). Springer International Publishing

The body visualizer is just a visualization tool for a parametric 3D body model that provides metrically accurate anthropomorphic measurements based on laser scans of thousands of people from different ethnicities.

Response 9: We thank the reviewer for the suggestion of references. We have revised the paragraph and included references as follows (as mentioned above):

“Body Visualizer has used the dataset based on SizeUSA, American and European Surface Anthropometry Resource (CAESAR) as the basis for creating 3D body shapes visualization [28, 29]. It is a visualization tool for a parametric 3D body model that provides metrically accurate anthropomorphic measurements based on laser scans of thousands of people from different ethnicities. However, it is still lacking the 3D body models for pregnant and postpartum women especially for Asian women. Therefore, in this study we focused on simulating the 3D shape of pregnant and postpartum women for Thais.”

[28] Loper, M., Mahmood, N., Romero, J., Pons-Moll, G., & Black, M. J. (2015). SMPL: A skinned multi-person linear model. ACM transactions on graphics (TOG), 34(6), 1-16.

[29] Osman, A. A., Bolkart, T., & Black, M. J. (2020). Star: Sparse trained articulated human body regressor. In Computer Vision–ECCV 2020: 16th European Conference, Glasgow, UK, August 23–28, 2020, Proceedings, Part VI 16 (pp. 598-613). Springer International Publishing

Point 10: There are relevant studies on prediction of 3D body shape during pregnancy. I suggest the authors also have a look at this paper for example:

Balasubramanian, M., & Robinette, K. (2020). Longitudinal anthropometric changes of pregnant women: dynamics and prediction. International Journal of Fashion Design, Technology and Education, 13(3), 231-237.

Response 10: Thank you for your advice. We have added a sentence related to the study that the reviewer suggested as follows:

“There are some relevant studies on prediction of 3D body shape during pregnancy using multiple 3D body scans with a purpose of setting the standard sizing chart for maternity wear that addresses the changes throughout pregnancy [33].”

[33] Balasubramanian, M., & Robinette, K. (2020). Longitudinal anthropometric changes of pregnant women: dynamics and prediction. International Journal of Fashion Design, Technology and Education, 13(3), 231-237.

  1. Unclear methodology

Point 11: "3D body shape simulation of normal women, pregnant and postpartum women shape with body proportions are predicted in real-time and online from weight, height and gestational age." What do you mean by 'predicted in real-time'?

 Response 11: Thank you for your question. We have added the explanation about 'predicted in real-time' as follows:

“A real-time prediction in our study is a service that provides the predictions via an HTTP call to simulate the 3D shape of pregnant and post-partum women via the web browser after the users input their data.”

Point 12: The authors say they collected anthropomorphic data in a 6 months time span, however it is not clear how or who collected the data (was there a specialist measuring the women, did they do self-measurements, etc.) Were these repeated measurements? Were the same participants measured in time?

 Response 12: Thank the reviewer for pointing this out. We have explained more about our methodology in the manuscript as follows:

“For the data collection, it started at 12-16 weeks of pregnancy and 0 weeks of postpartum. The research assistant measured body circumferences of chest, waist, hip, thighs (left/right), and upper arms (left/right) and explained to the volunteers how to measure their body size by themselves. The measurement was delicated considering that the interrater held the measuring tape in the correct position and not too tight.

The measurement values could vary approximately +/- 2 cm. The measurements were taken at home by pregnant/postpartum women subjects with the assistance of someone at home. The measurement was carried every 4 weeks. There would be a reminder notice from the research assistant when the schedule was approaching. Three measurements were taken 3 at each position and the median value was recorded for each position. The volunteers sent their measured data to the research team via mobile LINE application each time they measured their body shape.”

Point 13: "All analysed data were 587 datasets, including 94 women who were 12-week gestation; 98 of 16-week gestation; 91 of 20-week gestation; 82 of 24-week gestation; 79 of 28-week gestation; 78 of 32-week gestation and 65 of 36-week gestation" This information is hard to understand.

 Response 13: Thank you for your comment. We have revised this paragraph as follows:

“The data from 98 pregnant women volunteers were collected and analyzed. Information of all pregnant volunteers is shown in Table 1. The data were 587 sets in total. Each data contained woman's age, pre-pregnancy weight, height, gestational age, weight gain during pregnancy, inseam (measure once at 12–16 week pregnancy), and body circumference measurements: chest, waist, hip, upper arm (left/right), and thigh (left/right). The data used for this study were from 94 women who were 12-week gestation; 98 of 16-week gestation; 91 of 20-week gestation; 82 of 24-week gestation; 79 of 28-week gestation; 78 of 32-week gestation and 65 of 36-week gestation, a total of 587 sets.”

Point 14: In order to create the 'pregnant body shape' the authors utilize a combination of the following 'bodies': "Details of creating a 3D normal female shape by combining thin, big breast, big waist, big hip, tall and long legs avatars". It is unclear why this choice was made, or what supports making such a mix of 'attributes'/'3D body shapes'.

Response 14: To clarify the reviewer’s comments regarding the creation of the pregnant body shape, we have explained more in the manuscript as follows:

“The idea of utilizing a combination of the avatar bodies for 3D shape simulation came from the morphing technique [50]. Morphing is a geometric interpolation technique, which has to mix different characteristics of the objects. The body shape simulation by adjusting only a specific part was a challenge. For example, chest or hip circumference can be set bigger or smaller with the least impact on the waist and others. Therefore, our team designed the avatars in different ways to combine the shape of the body and the users are able to adjust the size of specific parts as needed. Therefore, the 3D non-pregnant female shape was created by combining thin, big breast, big waist, big hip, tall and long legs avatars using the morphing technique to make it easier to adjust only specific part of the body.”

Reference

[50] Kang J.Y., Lee B.S., Application of morphing technique with mesh-merging in rapid hull form generation, International Journal of Naval Architecture and Ocean Engineering, Volume 4, Issue 3, 2012, Pages 228-240, DOI: 10.2478/IJNAOE-2013-0092.

Point 15: On page 6, the authors write: "From the experiment, it was found that the K0 - K5 values were also proportional to the body mass index (BMI) and also depended on the gestational age as well." Here, for the first time, a 'experiment' is mentioned but there is no explanation of it in the paper.

 Response 15: To clarify the reviewer’s comment, we have explained more in the manuscript as follows:

“The simulation of non-pregnant body shape from our previous study [42], showed that the variables K_1- K_5 depended upon the BMI values. Therefore, a similar experiment was performed in this study by testing on 30 pregnant female subjects whose 3D data were simulated using equation 2 in comparison with the statical measurement from equation 1.”

In our experiment, the K0-K5 values are the sum between K00-K05 and the corresponding between Alp0-Alp5 shown in equations (3):

The values of Chest, Waist, Hip, and Inseam which are the values that defined K00-K05 were calculated from non-pregnant female body shape according to the article by Sinthanayothin et al. [42] which can be expressed by linear equations shown in equation (4).

From our experiments of cross-sectioning and measuring the circumference of 3D simulation figures, the Alp0 – Alp5 were functions of the BMI which could be calculated as the following morphing equations. The quadratic functions derived from second-order polynomial regression and parameters from the experiment were performed to obtain a 3D pregnant woman model that was the closest to the calculated statistical value as shown in equation (5).

[42]: Sinthanayothin C, Bholsithi W, Gansawat D, et al. Simulation of three-dimensional female body shapes with proportional representation for various weights and heights. SIMULATION. 2020;96(11):851-866. DOI:10.1177/0037549720944466

Point 16: Multiple equations contain constant values (e.g. equation 3), however, the rationale behind these constants is not explained in the paper.

Response 16: Regarding the constant values in equation 3, 13 values came from non-pregnant women simulation. Because the simulation of pregnant and postpartum women is modified based on the nonpregnant / non-postpartum female from the previous study.

Therefore, we have explained about all of the constants in equation 3 in the manuscript as follows:

“The simulations of pregnant and postpartum women were further modified based on data from non-pregnant female from our previous study [42]. The constant values of equation 3 were derived from the size of the designed avatars as mentioned in [42]. “The thin avatar was used as a default or initial model with the minimum values of chest, waist, and hip of approximately 57, 40, and 68 cm, respectively. The avatar with the big breast was applied to adjust the size of the chest values. The chest circumference of the avatar with the big breast was set as maximum chest values of approximately 200 cm. Similarly, for the avatar with the large waist and with the big hip, the size of the waist and hip of these avatars were set as maximum values of approximately 160 and 180 cm, respectively. For inseam, the default value for the thin avatar was approximately 73 cm. In this work, the minimum and maximum values of the inseam have been set to 48 and 120 cm, respectively. The last avatar (tall avatar) with the height of 200 cm is set to be the maximum height value.”

[42]: “Sinthanayothin C, Bholsithi W, Gansawat D, et al. Simulation of three-dimensional female body shapes with proportional representation for various weights and heights. SIMULATION. 2020;96(11):851-866. DOI:10.1177/0037549720944466”

Point 17: Authors make a differentiation in four categories of BMI:

"The 3D simulation of pregnant women can simulate at various gestational ages for women with four types of pre-pregnancy BMI: Underweight (BMI <18.5), Normal weight 256 (18.5 ≤ BMI ≤ 24.9), Overweight (25.0 ≤ BMI ≤ 29.9) and Obese (BMI ≥ 30.0)"

However, to my understanding, there is no factor in the equations to account for these categories. Why is this differentiation being made?

Response 17: We have explained more about the differentiation in four categories of BMI in Section 3.2 as follows:

Although there is no direct factor of BMI categories in our correlation analysis, the body proportions, chest, waist, hip, thigh, and upper arm circumferences were calculated using multiple linear regression method based on 587 data collected from 98 pregnant women. However, when a user wanted to predict her 3D pregnancy shape at other gestation ages using the web app (Z-Size Ladies), weight gain during pregnancy was unknown. Therefore, weight gain during pregnancy (Wg) would be predicted from pre-pregnancy BMI as shown in Table 2 based on the Institute of Medicine (IOM), 2009 [46].

[46]: Institute of Medicine, US.; National Research Council, US. Summary. In Weight Gain During Pregnancy: Reexamining the Guidelines., 1st ed.; Rasmussen, K.M., Yaktine, A.L., Eds.; National Academies Press: Washington D.C., USA, November 2009; DOI: 10.17226/12584, pp. S-2.

We also explained in the Discussion as follows:

Different types of pre-pregnancy BMI indicates the differences in weight gain during pregnancy. Therefore, 3D simulation of pregnant women was simulated at various gestational ages for women with four types of pre-pregnancy BMI according to IOM 2009 [46].  

Point 18: "The data of 83 postpartum women volunteers were collected and analysed". Are these some of the women who were part of the pregnancy group?

Response 18: The data of 83 postpartum women volunteers that were collected and analyzed, were from a completely different group from the pregnancy group. In addition, the volunteers (75 pregnant and 74 postpartum women) for testing the simulation application were also a new group.

Point 19: "The postpartum weight Wppt was predicted from the review articles". Why not from observations in your post-pregnancy group?

Response 19: Thank you for your comment, we have explained more about the postpartum weight Wppt in the manuscript as follows:

“The postpartum weight (Wpp) from measurement was already used as an independent variable in calculating the body circumference of postpartum women according to Equation 7. However, when a user wants to predict her 3D postpartum shape at other postpartum weeks using web app (Z-Size Ladies), postpartum weight is unknown. In the case of calculating the postpartum weight as a dependent variable, it would be complicated since it involved many factors such as BMI, pre-pregnancy weight, gestational weight gain, baby weight, and postpartum age. Moreover, data must be divided into 4 groups according to BMI types (underweight, normal weight, overweight and obese). Therefore, data from 83 postpartum women must also be divided into 4 groups, resulting in less than 30 postpartum women in each group. Data with n < 30 may not be sufficient for statistical analysis calculations [51].

Therefore, in the postpartum simulation application, the postpartum weight (Wpp) was predicted from the review articles.”

[51] Sharma A. Is n = 30 really enough? A popular inductive fallacy among data analysts. 2020. Available online at: https://towardsdatascience.com/is-n-30-really-enough-a-popular-inductive-fallacy-among-data-analysts-95661669dd98

Point 20: The accuracy is reported by means of correlation coefficients, it would be in my opinion clearer to report also absolute errors, and in general a more thorough analysis would be helpful.

Response 20: Following the reviewer’s suggestion, we have added the relative error (percentage of absolute error) in the accuracy reported in the manuscript.

Point 21: The 'satisfaction' test is fine, however, at the beginning of the paper, the authors claim that their simulation will: (1) raise awareness in women, (2) encourage women to prevent overweight before and during pregnancy, (3) maintain long term health. None of these outcomes have been measured, thus, making it not possible to evaluate the actual success of the study (if these points are really the goal of the study).

Response 21: We agree with the reviewer. Therefore, we revised our objectives to 1. Develop a simulation tool of 3D body shape for pregnant and postpartum women; 2. Test the accuracy of the developed body simulation tool. We revised the objectives in the manuscript as follows:

“Application Z-Size Ladies [31] described in this paper was intended to collect user data in the form of BMI timeline and simulate the 3D female body shape for non-pregnant, pregnant women, and postpartum women. Z-Size Ladies application [31] is a tool that helps pregnant and postpartum women to simulate their 3D-body shapes. The aim of developing the Z-Size Ladies application is to be a tool that can create a precise online 3D model for non-pregnant, pregnant, and postpartum women in real-time with a simple set of input data of their weight, height, and gestational age. The tool was validated by several linear regression studies and users survey. The simulation of the body shape from body measurements can be considered as a low-cost alternative to full-body 3D scanning [32].”

References:

[31] Z-Size Ladies: Web Application for BMI Timeline & Program assesses ladies’ sizes and facial images. 2021. Available at: https://zsize.openservice.in.th/

[32] Gallucci A., Znamenskiy D., and Petkovic M. Prediction of 3D Body Parts from Face Shape and Anthropometric Measurements. Journal of Image and Graphics, Vol. 8, No. 3, September 2020. DOI: 10.18178/joig.8.3.67-74

  1. Wording

Point 22: The term "normal women" to describe women who are not pregnant or in the "post-partum" period is problematic.

Response 22: We have revised the word ‘normal women’ to ‘non-pregnant women’ throughout the manuscript.

Point 23: The authors use the word "datasets" throughout their manuscript, in a way that I assume means just data or data points. e.g. "They used datasets from 25 pregnant Caucasian ladies". This should be revisited and corrected.

Response 23: We have revised the word ‘datasets’ to ‘data’ throughout the manuscript.

Point 24: In equation 6, gestational age is represented by the variable 'NV' but earlier in the paper (e.g. equation 1) it is represented by the variable Wkp.

Response 24: We have revised the variable ‘NV’ in equation 6 to ‘Wkp’.

Point 25: A general major issue is the lack of clarity in the general writing, structure and flow of the paper. It is difficult to fully follow the logic of the authors.

Response 25: We thank the reviewer again for your insightful review of our work. We have revised the manuscript accordingly to the reviewer's suggestions with more details and support references. We hope that the revised manuscript is clearer and easier to follow.

Reviewer 4 Report

I wish to thank the authors for the opportunity to review their work.

This article presents a tool that helps pregnant women to simulate their 3D-body shapes, basing on height, weight and gestational age. The tool was validated by users by means of a survey, and several linear regression studies seem to have been made.

The article is interesting and original, and presentation is ok, but the authors do not explain which statistical tools they are using for the study, and therefore the scientific soundness is below-average. I would recommend to make several important changes to it before it could be published:

General comments:

  • English should be deeply revised by a native English speaker, it is sometimes difficult to understand some parts of the manuscript.
  • More information about how the tool was developped should be included, it remains unclear how the depictions of bodies were created and shown on the site, how was the application programmed, etc.
  • Statistical study needs to be better explained, indicating which statistical software was used, how were correlations calculated, which were the confidence intervals, etc.

Specific comments:

L6-21- All authors seem to have the same affiliation, I would recommend to indicate it just by a single number (1). Also, usually only corresponding author e-mail is included, but if the authors want to include every e-mail address, I would recommend to put them at the end of the line in the same order as the authors appear above.

L24 – BMI has not been defined before, please indicate the meaning of every abbreviation first time it appears.

L40 – A citation would be needed to support such a strong statement.

L41-43 – Those reports should be cited.

L50-52 – A citation would be needed to support this statement.

L64 – SizeThailand term should be better explained . Is it an Institution, a NGA, a website…?. Same with SizeUSA in L76

L82-83 – “Z-Size Ladies v.1 database” should be cited or better explained.

L96 – MRI acronym should be explained the first time it appears.

L208 – “Turbosquid” reference should be cited or better explained.

Author Response

Response to Reviewer 4 Comments

Point 1: I wish to thank the authors for the opportunity to review their work.

This article presents a tool that helps pregnant women to simulate their 3D-body shapes, basing on height, weight and gestational age. The tool was validated by users by means of a survey, and several linear regression studies seem to have been made.

The article is interesting and original, and presentation is ok, but the authors do not explain which statistical tools they are using for the study, and therefore the scientific soundness is below-average. I would recommend to make several important changes to it before it could be published:

Response 1: We would like to thank the reviewer for your kind words and encouragement. We have provided more details on the statistical tool and updated several important changes to the manuscript accordingly to your comments.

General comments:

Point 2: English should be deeply revised by a native English speaker, it is sometimes difficult to understand some parts of the manuscript.

Response 2: Thanks for your advice, we will submit the manuscript to editing services (MDPI Author Services) for revisions by native English-speaker in minor revision after completing major revision correction.

Point 3: More information about how the tool was developped should be included, it remains unclear how the depictions of bodies were created and shown on the site, how was the application programmed, etc.

Response 3:  Thanks for your comment, we have explained more about how the proposed tool was developed in the methodology section as follows:

“The real-time visualization of 3D morphing of pregnant and postpartum female body shapes on the online Z-Size Ladies web application was implemented using the three.js library [47] incorporated with HTML5, JavaScript, and CSS for client-side development. Python, flask, and MySQL were employed for the server-side. Three.js was used because it was a cross-browser JavaScript library to ease the process of creating and displaying real-time 3D computer graphics and animation in the web browser.”

“The idea of utilizing a combination of the avatar bodies for 3D shape simulation came from the morphing technique [50]. Morphing is a geometric interpolation technique, which mixed different characteristics of the objects. The body shape simulation that adjusted only a specific part was a challenge. For example, chest or hip circumference could be set bigger or smaller with the least impact on the waist and others. Therefore, our team designed the avatars in different ways to combine the shape of the body and to be able to adjust the size of specific parts as needed. Therefore, the 3D normal female shape was created by combining thin, big breast, big waist, big hip, tall and long legs avatars using the morphing technique to make it easier to adjust only specific part of the body.”

Reference

[47] : Three JS, 2018.  Available online at:   https://threejs.org/

[50] Kang J.Y., Lee B.S., Application of morphing technique with mesh-merging in rapid hull form generation, International Journal of Naval Architecture and Ocean Engineering, Volume 4, Issue 3, 2012, Pages 228-240, DOI: 10.2478/IJNAOE-2013-0092.

Point 4: Statistical study needs to be better explained, indicating which statistical software was used, how were correlations calculated, which were the confidence intervals, etc.

Response 4: The short explanation of the statical study and tools used were mentioned with a short explanation of how correlations were calculated in Section 3.2.

“Wendland et al. [43] investigated the relationship between waist circumference and obesity-related pregnancy. The variables used in the correlation analysis were age, height, gravida, gestational age, uterine height, gestational BMI, and pre-pregnancy BMI. Similar work by Ricalde et al. [44], reported that some postpartum women's anthropometric was related to birth weight. Therefore, in this pregnancy study, the relationships between the body shape proportion and variables of pregnancy such as woman's age, weight, height, gestational age, weight gain during pregnancy were analyzed using multiple linear regression which could be calculated in Excel [45] as shown in equation (1).”

[43] Wendland E.M., Duncan B.B., Mengue S.S., Nucci L.B., & Schmidt M. (2007). Waist circumference in the prediction of obesity-related adverse pregnancy outcomes. Cadernos de saude publica, 23 2, 391-8. DOI:10.1590/S0102-311X2007000200015

[44] Ricalde, A.E., Velásquez-Melendez, G., Tanaka, A.C., & de Siqueira, A.F. (1998). Mid-upper arm circumference in pregnant women and its relation to birth weight. Revista de saude publica, 32 2, 112-7. DOI:10.1590/S0034-89101998000200002

[45]  Cameron A.C. EXCEL 2007: Multiple Regression, 2007. Available online at http://cameron.econ.ucdavis.edu/excel/ex61multipleregression.html

Table 11-14, 95% confidence interval (lower and upper) and the relative error in the 3rd-5th rows were added. A plot of confidence intervals chart was also shown in Figure 7-10.

Specific comments:

Point 5: L6-21- All authors seem to have the same affiliation, I would recommend to indicate it just by a single number (1). Also, usually only corresponding author e-mail is included, but if the authors want to include every e-mail address, I would recommend to put them at the end of the line in the same order as the authors appear above.

Response 5: The affiliations have been revised as follows.

Chanjira Sinthanayothin 1*, Piyanut Xuto 2, Wisarut Bholsithi 1, Duangrat Gansawat 1,                     Nonlapas Wongwaen 1, Nantaporn Ratisoontorn 1, Parut Bunporn 1 and Supiya Charoensiriwath 1

1, National Electronics and Computer Technology Center, National Science and Technology Development Agency, Pathumthani 12120, Thailand; chanjira.sinthanayothin@nectec.or.th, wisarut.bholsithi@nectec.or.th, duangrat.gansawat@nectec.or.th, nonlapas.wongwaen@nectec.or.th, nantaporn.ratisoontorn@nectec.or.th, parut.bunporn@nectec.or.th, supiya.charoensiriwath@nectec.or.th  

2    Faculty of Nursing, Chiangmai University, Chiangmai, Thailand; piyanut.x@cmu.ac.th

*   Correspondence: chanjira.sinthanayothin@nectec.or.th

Point 6: L24 – BMI has not been defined before, please indicate the meaning of every abbreviation first time it appears.

Response 6: We have defined BMI and other abbreviations the first time it appears.

Point 7: L40 – A citation would be needed to support such a strong statement.

Response 7: We revised the sentence and added citations following the reviewer’s suggestion:

“Obesity during pregnancy is a serious health problem for women. Worldwide, obstetricians and midwives have confronted an increase in obesity among pregnant women [1, 2].”

[1] Poston, L., Harthoorn, L., van der Beek, E. et al. Obesity in Pregnancy: Implications for the Mother and Lifelong Health of the Child. A Consensus Statement. Pediatr Res 69, 175–180 (2011). DOI: 10.1203/PDR.0b013e3182055ede

[2] Chen C, Xu X, Yan Y. Estimated global overweight and obesity burden in pregnant women based on panel data model. PLoS One. 2018 Aug 9;13(8):e0202183. DOI: 10.1371/journal.pone.0202183.

Point 8: L41-43 – Those reports should be cited.

Response 8: Thank you for your comment, those reports were cited as follows:

“Reports [3, 4, 5] showed that women with a pre-pregnant body mass index (BMI) of either overweight or obese levels are at risk of developing diabetes during pregnancy compared to women with normal pre-pregnant BMI, even after taking the weight gains during a normal pregnancy into account.”

References:

[3] Black K.I., Schneuer F., Gordon A., Ross G.P., Mackie A., Nassar N. Estimating the impact of change in pre-pregnancy body mass index on development of Gestational Diabetes Mellitus: An Australian population-based cohort. Women and Birth, 2022, DOI: 10.1016/j.wombi.2021.12.007.

[4] Persson M, Pasupathy D, Hanson U, et al. Pre-pregnancy body mass index and the risk of adverse outcome in type 1 diabetic pregnancies: a population-based cohort study. BMJ Open 2012;2:e000601. DOI: 10.1136/bmjopen-2011-000601

[5] Heude B, Thiébaugeorges O, Goua V, et al. Pre-pregnancy body mass index and weight gain during pregnancy: relations with gestational diabetes and hypertension, and birth outcomes. Matern Child Health J. 2012;16(2):355-363. DOI:10.1007/s10995-011-0741-9

Point 9: L50-52 – A citation would be needed to support this statement.

Response 9: We have updated the sentence with the citations to support as follows:

“It was reported that if a woman in the postpartum period was unable to regulate her weight to her pre-pregnant weight within six months, postpartum weight retention could predict future weight gain and long-term obesity [10]. Another study suggested that the BMI of a woman of more than six months postpartum would indicate the retaining extra body fluids produced during pregnancy, as well as extra fat during the first six months postpartum [11].”

[10] van der Pligt P, Willcox J, Hesketh KD, Ball K, Wilkinson S, Crawford D, Campbell K. Systematic review of lifestyle interventions to limit postpartum weight retention: implications for future opportunities to prevent maternal overweight and obesity following childbirth. Obes Rev. 2013 Oct;14(10):792-805. DOI: 10.1111/obr.12053.

[11] Arizona WIC (Women Infant and Children) Nutrition Care Guidelines: Breastfeeding and Postpartum Women, 2015. Available online at: https://azdhs.gov/documents/prevention/azwic/face-to-face/2015/jan/6-Breastfeeding-and-Postpartum-Women.pdf

Point 10: L64 – SizeThailand term should be better explained . Is it an Institution, a NGA, a website…?. Same with SizeUSA in L76

Response 10: We have explained about SizeThailand with the citation.

“Then data were analysed and processed to create simulation of 3D modelling of pregnant and postpartum women based on data of non-pregnant female shapes from SizeThailand [19, 20].”

“SizeThailand is a national sizing surveys project among 13,442 adults, both males, and females across Thailand at various ages [30].”

References

[19] Wells, J., Treleaven, P. & Charoensiriwath, S. Body shape by 3-D photonic scanning in Thai and UK adults: comparison of national sizing surveys. Int J Obes 36, 148–154 (2012). DOI: 10.1038/ijo.2011.51

[20] Wells JC, Charoensiriwath S, Treleaven P. Reproduction, aging, and body shape by three-dimensional photonic scanning in Thai men and women. Am J Hum Biol. 2011 May-Jun;23(3):291-8. DOI: 10.1002/ajhb.21151.

[30] SizeThailand (In Thai). Available online at  http://www.sizethailand.org/

Point 11: L82-83 – “Z-Size Ladies v.1 database” should be cited or better explained.

Response 11: We have revised the paragraph as follows:

“The body shape proportion of pregnancy and postpartum women were analysed using linear regression of 587 pregnancy data and 503 postpartum data. The simulations of pregnant and postpartum women were further modified based on non-pregnant simulation from our previous study which was analysed from the SizeThailand database with 6,767 females’ data [19, 20].”

References

[19] Wells, J., Treleaven, P. & Charoensiriwath, S. Body shape by 3-D photonic scanning in Thai and UK adults: comparison of national sizing surveys. Int J Obes 36, 148–154 (2012). DOI: 10.1038/ijo.2011.51

[20] Wells JC, Charoensiriwath S, Treleaven P. Reproduction, aging, and body shape by three-dimensional photonic scanning in Thai men and women. Am J Hum Biol. 2011 May-Jun;23(3):291-8. DOI: 10.1002/ajhb.21151.

Point 12: L96 – MRI acronym should be explained the first time it appears.

Response 12: Magnetic Resonance Imaging (MRI) has been added to the MRI acronym.

Point 13: L208 – “Turbosquid” reference should be cited or better explained.

Response 13: The term of Turbosquid is explained as follows:

“TurboSquid is a digital media company that sells 3D models used in 3D graphics to a variety of industries, including computer games, architecture, and interactive training [48, 49].”

Reference

[48] Turbosquid: Pregnant Woman by MotionCow. Available from:    https://www.turbosquid.com/3d-models/pregnant-woman-pregnancy-3d-max/693351 [Accessed 15th December 2018].

[49] Turbosquid. 3D Models for Professionals. Available from:    https://www.turbosquid.com/ [Accessed 15th December 2018].

We have carefully consolidated the manuscript and sent it to the Professional Authorship Center, Thailand National Science and Technology Development Agency (NSTDA) for review and revision.

Round 2

Reviewer 2 Report

Authors have considered all the concers adequately and hence the manuscript can be accepted. 

Reviewer 4 Report

The authors have addressed all of my concerns.